# Postmitotic differentiation of human monocytes requires cohesin-structured chromatin

Julia Minderjahn[1,4], Alexander Fischer [1], Konstantin Maier[1], Karina Mendes[1,5], Margit Nuetzel[1], Johanna Raithel[2], Hanna Stanewsky[1], Ute Ackermann[1], Robert Månsson[3], Claudia Gebhard[2] & Michael Rehli [1,2✉]

Cohesin is a major structural component of mammalian genomes and is required to maintain loop structures. While acute depletion in short-term culture models suggests a limited importance of cohesin for steady-state transcriptional circuits, long-term studies are hampered by essential functions of cohesin during replication. Here, we study genome architecture in a postmitotic differentiation setting, the differentiation of human blood monocytes (MO). We profile and compare epigenetic, transcriptome and 3D conformation landscapes during MO differentiation (either into dendritic cells or macrophages) across the genome and detect numerous architectural changes, ranging from higher level compartments down to chromatin loops. Changes in loop structures correlate with cohesin-binding, as well as epigenetic and transcriptional changes during differentiation. Functional studies show that the siRNA-mediated depletion of cohesin (and to a lesser extent also CTCF) markedly disturbs loop structures and dysregulates genes and enhancers that are primarily regulated during normal MO differentiation. In addition, gene activation programs in cohesin-depleted MO-derived macrophages are disturbed. Our findings implicate an essential function of cohesin in controlling long-term, differentiation- and activation-associated gene expression programs.

[1] Department of Internal Medicine III, University Hospital Regensburg, 93053 Regensburg, Germany. [2] Leibniz Institute for Immunotherapy, c/o University Hospital Regensburg, 93053 Regensburg, Germany. [3] Center for Hematology and Regenerative Medicine Huddinge, Karolinska Institutet, Stockholm, Sweden. [4] Present address: Sandoz GmbH, Biochemiestraße 10, 6336 Langkampfen, Austria. [5] Present address: Universidade Católica Portuguesa, Center for Interdisciplinary Research in Health (CIIS), Institute of Health Sciences (ICS), Viseu, Portugal. ✉email: michael.rehli@ukr.de

The ring-shaped protein complex cohesin and the CCCTC-binding factor CTCF comprise a core architectural unit of mammalian genomes. They colocalize at DNA loop anchors[1,2] and form the boundaries of larger contact domains that spatially separate genes and regulatory elements from each other[3–5]. Current models suggest a dynamic, cyclic process of cohesin-dependent loop formation, which involves the NIPBL-mediated loading of cohesin to DNA, and the extrusion of loops (confined by CTCF-bound sites in convergent orientation), and the WAPL-mediated release of cohesin[6]. Cohesin-dependent DNA loop formation and CTCF-mediated insulation of contact domains are generally considered important for regulating the interplay of promoters and enhancers during gene transcription[7,8].

While the functional importance of DNA loops and their anchors is well described for individual gene loci[4,9–11], their relevance for global transcription control is less well established. Recent studies reported the rapid decay of contact domains as an immediate consequence of acute CTCF or cohesin removal but failed to observe major transcriptional changes[12–14], questioning the importance of spatial genome organization for maintaining transcriptional output. However, it is unclear whether these observations, which were made under steady-state conditions, also reflect the relevance of 3-dimensional genome organization in the context of developmental, differentiation, or activated gene expression programs. It has been noted that genes with dynamic transcription (as observed in activation or differentiation processes) are associated with higher contact frequencies as compared to housekeeping genes[15]. This type of genes may also be more sensitive to cohesin-depletion[16,17], highlighting the need for additional models to clarify the relationship between cohesin/CTCF function and transcription regulation.

However, due to the importance of cohesin for cell cycle progression[18], studying the relevance of cohesin-dependent loop formation in the more complex settings of differentiation/development and over longer time periods remains difficult. One suitable model that allows the observation of gene-regulatory changes over several days in the absence of cell division is the differentiation of human peripheral blood monocytes (MO). These innate immune cells normally develop from myeloid progenitors in the bone-marrow before they enter the bloodstream as mature and patrolling effector cells[19]. They present a typical horseshoe-shaped nucleus, which rapidly reshapes upon differentiation into MO-derived cell types like macrophages (MAC) or dendritic cells (moDC). In vitro, this differentiation is accompanied by abundant changes in chromatin accessibility, transcription, or epigenetic landscapes, which all proceed in the absence of proliferation[20,21]. Notably, mutations in cohesin complex members, including RAD21, have been identified as early events in the pathogenesis of myeloid neoplasms[22,23] and cohesin mutations or knockdown of cohesin subunits were shown to impair hematopoietic differentiation and enforce stem cell programs in hematopoietic progenitor cells[16,24–26].

Here, we use this naturally post-proliferative primary cell system to study the function of CTCF and cohesin in differentiation. We generate high-resolution in situ Hi-C maps ($>700\times10^6$ interactions per cell type) for human MO, MO-derived macrophages (MAC), and dendritic cells (moDC), and compare the effects of cohesin or CTCF depletion in each model. The present work provides evidence for abundant differentiation-associated changes in the genome organization of these cells and shows that the genes and enhancers affected by cohesin depletion are to a large extent also regulated during normal differentiation. We also show that the functional repertoire of MAC is altered when cohesin is depleted during differentiation. Hence, cohesin is required for the proper execution of differentiation- and activation-associated transcription programs.

## Results

**Genome-wide changes during MO differentiation.** To study the relationship between genome and transcription regulation, we analyzed two phenotypically distinct in vitro culture models for postmitotic human blood MO (Fig. 1a), including their differentiation into adherent, MO-derived macrophages (MAC, cultured for 7 days in the presence of 2% human AB-serum) or non-adherent MO-derived dendritic cells (moDC, cultured for 7 days in the presence of GM-CSF, IL-4 and 10% FCS). In MO and both differentiation endpoints, we initially mapped the global distribution of H3K27ac, a histone mark found at active regulatory elements (via ChIP-seq), accessible chromatin (via ATAC-seq), and transcription (via RNA-seq) for a total of three independent donors. As shown in Fig. 1b, the differentiation of MO into MAC or moDC induced abundant and consistent changes in chromatin accessibility, gene expression, and distribution of H3K27ac, both at the levels of individual peak regions or arrays of peak regions (super-enhancers). We observed characteristic gene expression changes, including the downregulation of known MO-specific genes (e.g. *CD300E*, *NR4A2* or *VCAN*), the upregulation of MAC-specific genes (e.g. *VSIG4*, *CYP19A1* or *CHIT1*) or moDC-specific genes (e.g. *CD1A*, *ALOX15* or *CCL13*) during differentiation, as shown exemplary in Fig. 1c and in more detail in a heatmap of the top variable genes across the three cell types (Supplementary Fig. 1a). Gene set enrichment analysis across published data sets confirmed cell identities (Supplementary Fig. 1b, c) and cell type-specific gene expression patterns coincided with the cell type-specific distribution of H3K27ac and accessible chromatin (Fig. 1d, e and Supplementary Fig. 1d, peak positions and results of differential gene or peak analysis are provided in Supplementary Data File 1). Both, known and *de novo*-derived motif signatures at cell type-specific, accessible chromatin or active regulatory regions (Fig. 1f and Supplementary Fig. 1e, f, respectively) as well as their co-association networks were typical for the three cell types, e.g. with EGR2 being induced during differentiation of MAC and moDC[20,27], STAT factors being active in moDC (as induced by GM-CSF and IL-4) and KLF signatures being lost in MAC[20] (Fig. 1f, g). In addition, differentiation of MO significantly altered the ranking of super-enhancers at several cell type-specifically expressed loci, as shown in Supplementary Fig. 1g, h. This includes e.g., the CIITA transcription factor, which drives the antigen-presentation program in moDC[28], *CISH*, which is required for moDC-mediated CTL activation[29], the Natural resistance-associated macrophage protein 1[30] (encoded by *SLC11A1*), which is involved in iron metabolism and host resistance to certain pathogens in MAC, or the ferredoxin gene locus (FDX1), which is involved in the biosynthesis of steroids[31], including Vitamin D (also see Supplementary Fig. 1g, h).

Having established that both differentiation endpoints showed typical features of MAC or moDC differentiation, we captured high resolution chromatin interactions in each of the three states (MO, MAC, moDC) using in situ Hi-C (between $760\times10^6$ and $930\times10^6$ interactions per cell type, quality metrics are shown in Supplementary Fig. 1i–k). In line with the differentiation-induced change in nuclear shape, we observed reproducible changes in interchromosomal interactions between MO and the two differentiated cell types, indicating global rearrangements of chromatin territories[32] during differentiation, as also previously observed in plasma cell differentiation[33]. Figure 1h shows regional differences in ratios of normalized interchromosomal contacts across the genome between MO and MAC. Changes

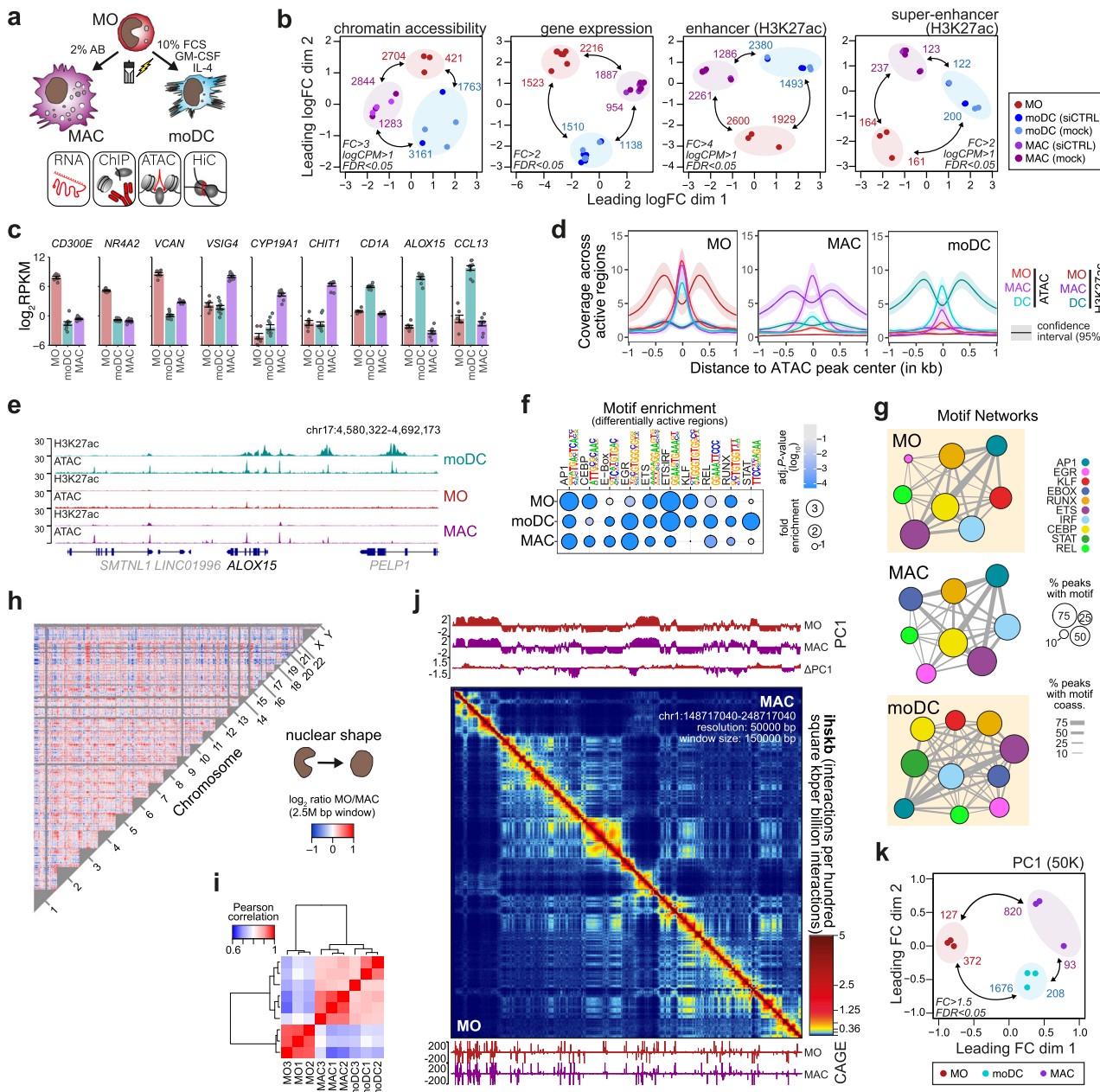

**Fig. 1 Epigenetic, transcriptional and architectural changes during MO differentiation. a** Schematic of the experimental setup. Since cells served as controls for the siRNA-mediated knockdown of structural proteins, MO were either mock-treated or electroporated with a control siRNA (siCTRL) after isolation and before culture. **b** Multidimensional Scaling (MDS) plots of ATAC-seq, RNA-seq, and H3K27ac ChIP-seq data sets. H3K27ac data are shown either on the levels of individual peak regions (enhancers) or arrays of peaks (super-enhancers). Numbers of differentially accessible sites, expressed genes, or H3K27ac defined enhancers or super-enhancers are indicated. **c** Bar plots showing expression levels (mean normalized RPKM ± SE based on the RNA-seq data; $n = 6$ control samples from three donors) of typical cell type-associated genes. **d** Histograms of ChIP- and ATAC-seq coverage at differentially active regulatory regions (based on their H3K27ac deposition), centered on overlapping open chromatin. **e** Genome browser tracks for a moDC-specific example region (additional examples in Supplementary Fig. 1d). **f** Motif enrichment (hypergeometric test, Benjamini-Hochberg (BH) corrected) across open chromatin regions shown in (**d**). **g** Motif co-association networks for regions given in (**d**). The size of each node represents the motif enrichment (fraction of peaks) and co-associated TF motifs are indicated by coloring. Edge thickness indicates the frequency of motif co-association. **h** Comparative interchromosomal contact map of MO and MAC. Coloring indicates the enrichment of contacts in MO (red) or MAC (blue) across 2.5 Mb windows. **i** Clustering of Pearson correlation values for all significant interchromosomal contacts between individual donors and cell types. **j** In situ Hi-C contact map of MO (lower left) and MAC (upper right) across a 50 Mb interval of chromosome 1. The map represents the average of 3 donor replicates per condition. Corresponding presentations for moDC are given in Supplementary Fig. 1l. Top, Tracks representing the first eigenvector values (PC1) of a principal component analysis (PCA) on a Hi-C correlation matrix at 50kB resolution (PC1), including a difference track as indicated. Bottom, CAGE-seq data. **k** MDS plot of PC1 differences between cell types. Numbers of significantly different 50 K windows are indicated. (**c**, **f**, **g**, **i**, **k**) Source data are provided as a Source Data file.

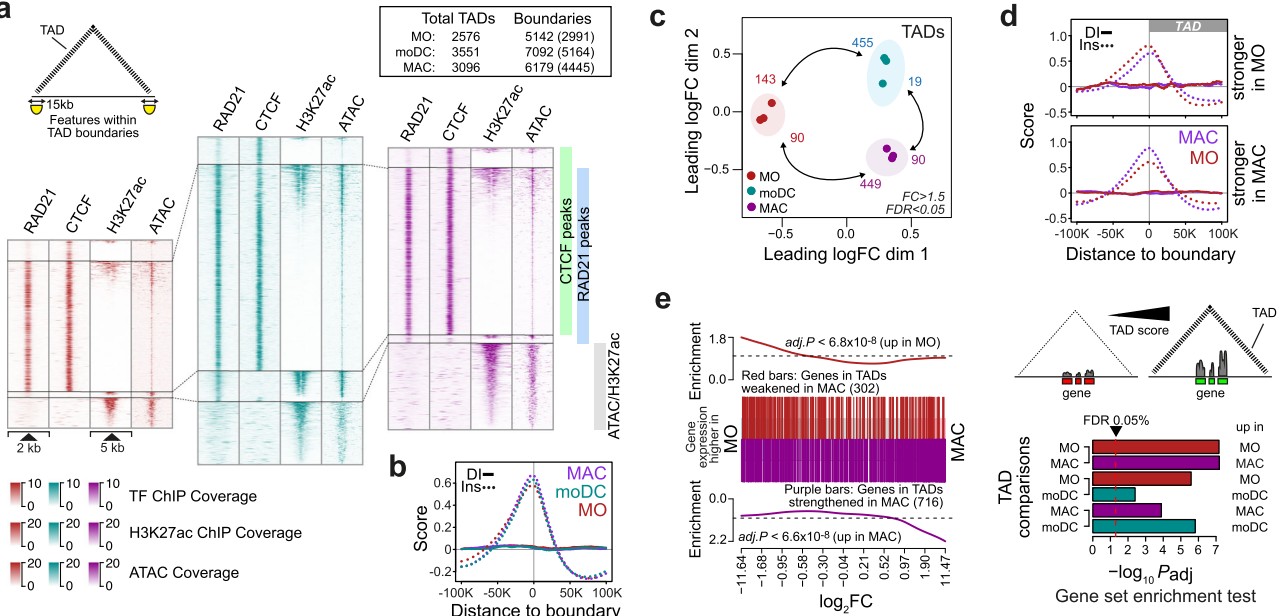

**Fig. 2 Changes in topology-associated domains during MO differentiation correlate with gene expression changes. a** Genomic distance distributions of RAD21, CTCF, H3K27ac ChIP-seq and ATAC-seq coverage at peak-centered boundaries of topology-associated domains (TADs) identified in MO, moDC and MAC. TAD boundary size is indicated in the schematic in the top left corner, where the triangular form indicates the TAD area. The numbers of TADs and non-overlapping boundaries are boxed in the upper right corner. The number of boundaries overlapping RAD21, CTCF, H3K27ac, or ATAC peaks are given in brackets. **b** Histograms of insulation scores (Ins, dotted lines) and directionality indices (DI, indicating preferences for contacts either upstream or downstream) across TAD boundaries in MO an MO-derived cells. **c** MDS plot comparing TAD score data sets of the indicated cell types. Numbers of significantly different TADs are indicated. **d** Histograms of insulation scores and directionality indices across differential TAD boundaries for MO versus MAC comparisons. **e** Gene set enrichment of differential TAD-associated genes. The enrichment of genes associated with cell type-specific TADs (MO in red, MAC in purple) across all genes ranked by their expression difference in MO versus MAC is shown exemplary in the barcode plot in the left panel. Results of gene set enrichment testing for all MO, MAC, and moDC comparisons are summarized in the bar plot shown in the bottom right panel. Shown are adjusted *P* values (two-sided rotation tests, BH correction for paired testing), indicating that changes in TAD strengths tend to associate with accordant changes in the expression of TAD-overlapping genes, as depicted in the schematic on top of the bar plot. (**b**, **d**, **e**) Source data are provided as a Source Data file.

were reproducible between donors, as indicated by the correlation matrix of significant interchromosomal contacts shown in Fig. 1i. Notably, interchromosomal contacts were more similar between moDC and MAC. An example map for intrachromosomal contacts is shown in Fig. 1j, indicating ample changes in chromatin compartments (indicated by the first eigenvector (PC1) values) between MO and MAC (corresponding MO/moDC and moDC/MAC comparisons are shown in Supplementary Fig. 1l; similarities between individual samples are visualized in Fig. 1k). As also observed on the interchromosomal level, PC1 changes between moDC and MAC were less pronounced (but still significant), suggesting that most higher-order compartment changes occur early during differentiation along both routes.

**Correlation of transcriptional and architectural changes**. We next tested whether MO differentiation was also associated with changes on the levels of chromatin loops and topology-associated domains (TADs). In total, we detected 2.6 K, 3.6 K, and 3.1 K TADs in MO, moDC, and MAC, respectively, which were frequently associated with co-binding of RAD21 and CTCF (Fig. 2a, positions of TADs and loops, as well as RAD21 and CTCF peak positions are provided in Supplementary Data File 2) at their boundaries (15k bp wide). Insulation scores slightly increased during differentiation (Fig. 2b), in line with previous observations in neuronal differentiation[34]. MO differentiation coincided with a general downregulation of cohesin components, particularly in MAC (Supplementary Fig. 2a). This also included cohesin unloading complexes, which may at least partially explain the

differentiation-associated increase in TAD boundary insulation, as previously suggested[34].

Cohesin-independent boundaries that were associated with H3K27ac or chromatin accessibility were enriched for typical cell type-specific enhancer (PU.1, CEBP, AP1) or general promoter motifs (NRF1, NFY; shown in Supplementary Fig. 2b), suggesting that regulatory elements contribute to TAD boundaries independent of CTCF or cohesin. Comparisons between cell types revealed the strengthening or de novo appearance of TADs during MO differentiation (Fig. 2c, results of differential TAD score analyzes are provided in Supplementary Data File 2), while fewer TADs appeared weakened. TAD boundaries with higher scores in MO or MAC showed accordant changes in insulation scores (Fig. 2d, similar plots for other comparisons are shown in Supplementary Fig. 2c, d).

To globally assess the relationships between architectural changes and gene expression, epigenetic features or transcription factor binding, we adopted gene set testing and asked whether features in a given feature list (e.g. genes or peaks overlapping differential loops or TADs) tend to be differentially expressed or active. Corresponding results for genes associated with TADs either strengthened or weakened during MO to MAC differentiation are presented in the right panel of Fig. 2e. The ranks of TAD-associated genes clearly indicate that TADs strengthened in MAC tend to contain genes that are upregulated in MAC, whereas TADs weakened in MAC tend to contain genes that are downregulated in MAC. Concordant tendencies were also detected in gene set enrichment analyzes of TADs that were different between MO and moDC, or moDC an MAC (adjusted *P*

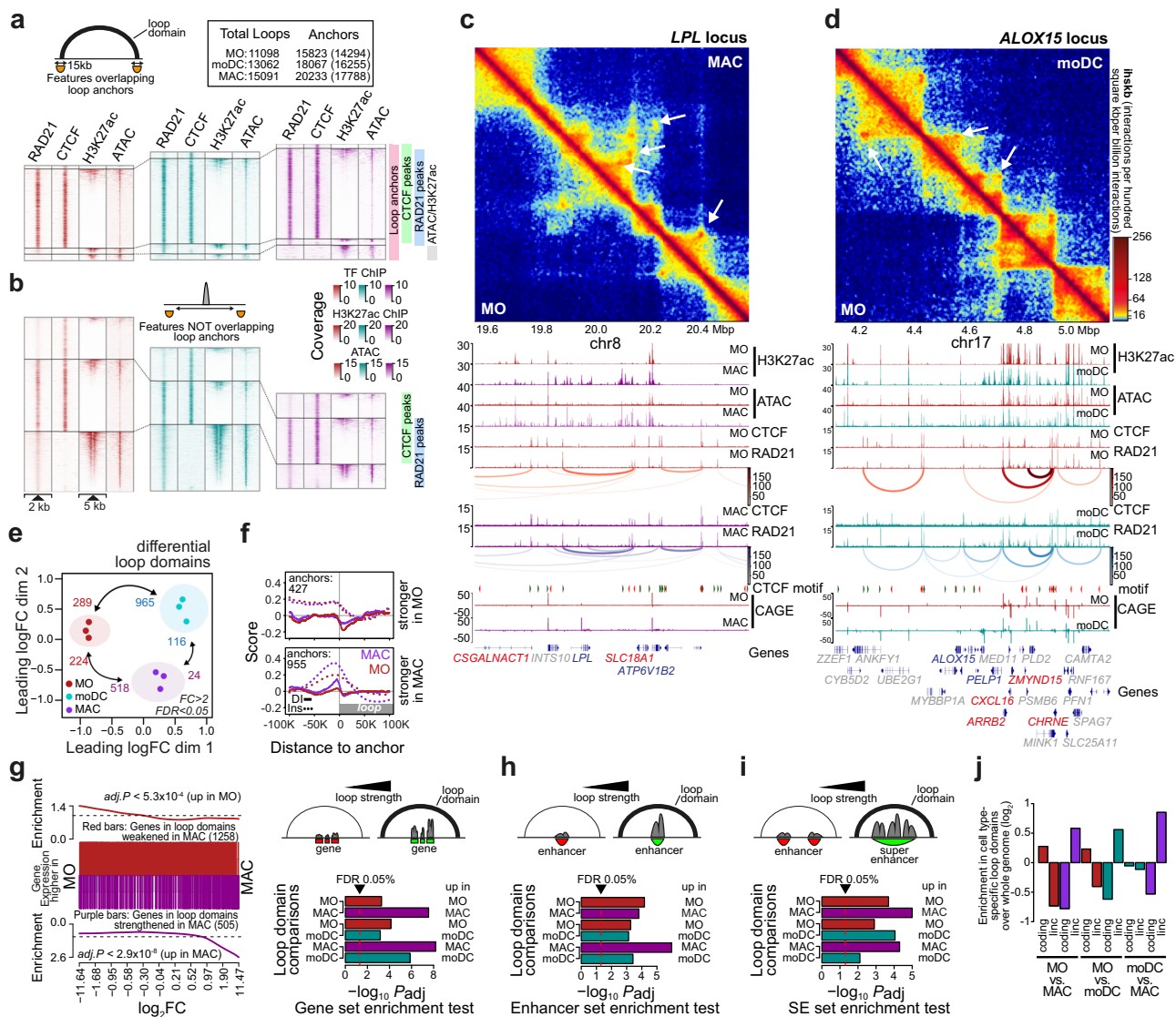

**Fig. 3 Architectural changes during MO differentiation correlate with gene expression changes. a, b** Genomic distance distributions of RAD21, CTCF, H3K27ac ChIP-seq and ATAC-seq coverage at peak-centered loop anchors (in a) or at CTCF/RAD21 peaks not overlapping anchors (in b). Loop anchor size is indicated in the schematic in the top left corner, where the arch indicates the loop domain. Numbers of loops and non-overlapping anchors are boxed in the upper right corner. The number of boundaries overlapping RAD21, CTCF H3K27ac or ATAC peaks are given in brackets. **c, d** Comparative in situ Hi-C interaction maps (at 10 kb resolution) and corresponding genome browser tracks for indicated example regions comparing MO and MAC or moDC. Tracks include ATAC-seq, ChIP-seq (H3K27ac, RAD21, CTCF) and CAGE-seq data as indicated. **e** MDS plot comparing loop score data sets of the indicated cell types. Numbers of significantly different loop domains are indicated. **f** Histograms of insulation scores and directionality indices across differential loop anchor regions in MO and MAC. **g** Gene set enrichment of loop-associated genes. Enrichment of genes associated with cell type-enriched loops (MO in red, MAC in purple) across all genes ranked by their expression in MO versus MAC (left panel). Right panel shows the summary of gene set enrichment testing for all MO, MAC, and moDC comparisons. Shown are adjusted P values (two-sided rotation tests, BH correction for paired tests), indicating that genes associated with stronger loops in a particular cell type tend to be higher expressed in the same cell type. Bars are colored according to cell types (MO in red, MAC in purple, moDC in turquoise). **h** Same as in (**g**) for enhancers (H3K27ac-marked regions). **i** Same as in (**g**) for super-enhancers. **j** Bar plot showing enrichment of LincRNA versus coding genes across cell type specific loop domains (coloring as in g). LincRNA genes are specifically enriched in MAC- and moDC-specific loops. (**f–j**) Source data are provided as a Source Data file.

enrichment P values are given in the right panel of Fig. 2e). Similar analyzes addressing H3K27ac (both on the level of individual peak regions or super-enhancers) showed that H3K27ac changed in the same direction as TAD scores (Supplementary Fig. 2e, f). Hence, changes in TADs generally correlated with concordant transcriptional or regulatory activity changes, which has also been noted in other systems[34,35].

On the level of loops, we detected 11.1 K, 13.1 K, and 15.1 K domains in MO, moDC, and MAC, respectively, which were mostly associated with co-binding of RAD21 and CTCF at their

anchor (Fig. 3a), and CTCF motifs were mostly in the preferred sense/antisense orientation (Supplementary Fig. 3a). We also plotted the same distributions across RAD21 and CTCF bound regions that were not associated with anchors of loop domains (Fig. 3b). The comparison of insulation scores between enhancers ranked by their activity, or subdivided by their association with loop anchors, RAD21, or CTCF showed that sites out-side of loop anchors also acted as insulators, suggesting that the in-situ Hi-C approach likely only detected a subset of loop domains. It also showed that insulation at loop anchors generally increased during

MO differentiation, even more pronounced as observed in TADs (Supplementary Fig. 3b). CTCF- and cohesin-independent enhancers at loop anchors showed specific motif signatures with YY1 being the top enriched motif (Supplementary Fig. 3c). This is in line with previous studies implicating this transcription factor in the formation of cohesin-independent loops[36].

Figure 3c, d show contact maps and genome browser tracks of example regions, which display gains or losses in contact loops (Additional examples are shown in Supplementary Fig. 3d). Numbers of significantly altered loop domains are given in the MDS plot shown in Fig. 3e, which also indicates that changes were similar across donors. Interestingly, loop anchors enriched in MO or MAC showed divergent distributions of insulation scores and directionality indices. While MAC-enriched loop anchors showed typical features of induced structural loops (increased insulation and directionality around the loop anchor), MO-enriched loop anchors showed similar insulation scores in both MO and MAC and only slight difference in directionality between MO and MAC, suggesting that the latter may not represent major structural changes (Fig. 3d). Similar differences were observed in moDC (Supplementary Fig. 3e, f).

To study the relationship between gene expression and architectural changes, we tested for the enrichment of genes associated with cell type-specifically strengthened loop domains in cell type-specifically expressed genes. As shown in Fig. 3g, genes associated with strengthened loop domains are also more highly expressed in the same cell type. The same type of relationship was also observed for H3K27ac-marked enhancer candidate regions (Fig. 3h) and super-enhancers (Fig. 3i). Hence, architectural changes during MO differentiation are frequently co-associated with corresponding epigenetic and transcriptional changes. Interestingly, loop domains that were strengthened or established during MO differentiation were enriched in non-coding genes (and depleted in coding genes, see Fig. 3j). However, this did not result in the detectable enrichment of lincRNA expression in MAC (or moDC) (Supplementary Fig. 3g), suggesting that de novo loop formation during MO differentiation is frequently initiated in chromosomal regions that are comparably poor in coding genes. In line with this, we observed a number of loops that were not associated with transcriptional changes but correlated with the appearance of novel regulatory elements (gain of accessibility/H3K27ac, e.g. the loop in the PKHD1L1 locus in Supplementary Fig. 3d marked with an asterisk and examples shown in Supplementary Fig. 3h). While the functional relevance of these loop domains is unclear, their generation suggests that structural changes during differentiation are induced by the generation of novel, differentiation-induced cohesin-loading sites.

**Properties of loop anchors of differentiation-associated loop domains**. We further studied the properties of loop anchors and correlated differential connectivity with signals for two major boundary factors, the cohesin complex component RAD21, and the transcription factor CTCF. Overlaps between cell type-enriched loop anchors and RAD21 or CTCF binding were strong in MAC or moDC and less pronounced in MO where strengthened loops more frequently lacked evidence for CTCF binding (Fig. 4a and Supplementary Fig. 4a, b). Motif enrichment analyzes across CTCF-independent anchor sites (which were often co-marked by H3K27ac) identified motifs corresponding to TFs that were previously also identified in enhancers in MO and MO-derived cells (like PU.1, AP1, and C/EBP, etc., see Fig. 4a, b and Supplementary Fig. 4a–c for other comparisons).

To test whether signals for RAD21 corresponded with altered loop strength, we performed peak set enrichment analyzes. As

shown for MO/MAC in Fig. 4c and in summary for all comparisons in Fig. 4d, the strengthening of loops during MO differentiation significantly correlated with higher signals of RAD21 peaks within corresponding loop anchors in the differentiated cell type. The weakening of differential loops during MO differentiation also showed a trend towards higher signals of RAD21 peaks in MO within corresponding anchors (summarized in Fig. 4d). Since we noted a substantial fraction of RAD21 sites not detected as loop anchors in Hi-C, we also analyzed motif signatures across differential cohesin signals (Supplementary Fig. 4d–f). Here, we also observed an enrichment of enhancer-associated TFs in MO, while in MAC, differential RAD21 sites were dominated by CTCF. ChIP-seq coverage for CTCF was less dynamic during MO differentiation and changes were less pronounced compared to RAD21 (Fig. 4a). However, we still observed a similar enrichment of anchor overlapping CTCF peaks in differentially bound peaks (Fig. 4e, f). Interestingly, the percentage of CTCF-bound sites and CTCF motif scores were both lower in MO-specific loop anchors compared to CTCF-bound sites in MAC-specific loop anchors, suggesting that lower affinity CTCF sites are lost during differentiation, while higher affinity CTCF sites are engaged in loop formation in MAC (Supplementary Fig. 4g). Overall, these findings suggest that MO derived cells lose regulatory loops but gain CTCF-dependent structural loops.

Supplementary Fig. 4h–k, show genome tracks covering example loci, including KLF4 and STK17B, which loose regulatory loops and down-regulate the expression of both genes during MO differentiation, as well as CYP19A1 and BTG2, where differentiation-induced loop structures correlate with either up- or down-regulation of genes contained within, suggesting that structural loops may not only foster, but also suppress gene regulation, e.g. by isolating genes from regulatory elements outside of the loop. Notably, the relationships between differentiation-associated TAD/loop changes and gene/enhancer activity changes were never uniform. As obvious from the bar code plots shown in Figs. 2e, 3g, 4g, i many genes or cis-elements were insensitive to changes in TAD or loop strength. This is in line with the mild effects on gene transcription observed after acute cohesin depletion[13].

Collectively, the analysis of architectural changes suggest that MO differentiation is associated with rearrangements in higher-order chromosomal territories (correlating with nuclear shape changes). We also observe differentiation-associated changes in TADs and loops that frequently parallel concordant changes in gene expression or enhancer/promoter activity. We also detect fewer CTCF-independent regulatory loops and a large gain of primarily CTCF-dependent structural loops during differentiation (schematically summarized in Fig. 4k).

**Architectural effects of cohesin or CTCF knockdown**. Our observations in the MO differentiation model suggested that alterations in genome structure correlated well with gene expression changes. To further explore the relationship between both types of changes and to determine, which types of genes required cohesin or CTCF during MO differentiation, we performed functional studies using siRNA-mediated knockdown of RAD21 and CTCF. The experimental setup is schematically depicted in Fig. 5a. To enable transfection of MO, we used specifically designed siRNAs with backbone-modifications that do not activate innate immune cells like human MO. As exemplary shown for MAC, protein levels of CTCF and RAD21 gradually decreased over the 7 day-culture period, reaching knockdown levels of 70–80% for CTCF and 80–90% for RAD21 at day 7 (Supplementary Fig. 5a). On transcript level the knockdown of

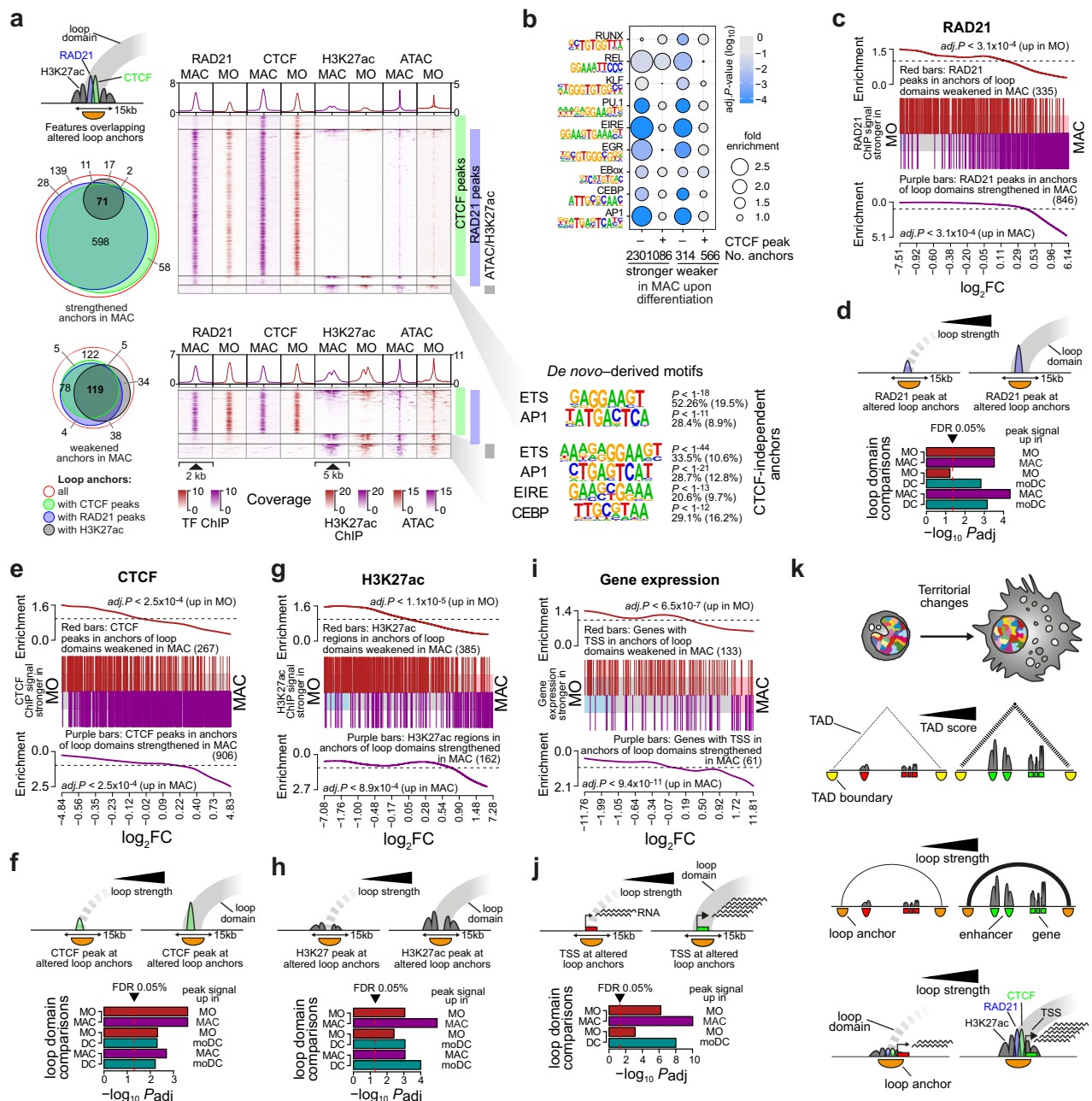

**Fig. 4 Differentiation-associated shift from regulatory to structural loops. a** Genomic distance distribution of RAD21, CTCF, H3K27ac ChIP-seq, and ATAC-seq coverage at peak-centered loop anchors (divided into four groups, as indicated, and sorted by H3K27ac coverage) that were either strengthened or weakened in MAC compared to MO. Loop anchor size is indicated in the schematic in the top left corner, where the arch indicates the loop domain. Venn diagrams on the left show the overlap between cell type-specific loop anchors, RAD21, CTCF, and H3K27ac peaks for each set. Top de novo-derived motifs for CTCF-independent loop anchors are given on the right along with the significance of motif enrichment (hypergeometric test) and the fraction of motifs in peaks (background values are in parenthesis). **b** Balloon plot showing motif enrichment along with color-coded P-values (hypergeometric test, BH adjusted) across open chromatin at CTCF-independent differential loop anchors or CTCF peak areas at CTCF-overlapping differential anchors in MAC compared to MO. **c** Peak set enrichment of loop anchor-associated RAD21 peaks (P-values: two-sided rotation tests, BH correction for multiple testing). The enrichment of peaks associated with cell type-specific loops (MO in red, MAC in purple) across all peaks ranked by their signal in MO versus MAC is plotted. **d** Summary of P-values (two-sided rotation tests, BH correction for paired tests) for pairwise comparisons between MO, MAC and moDC indicating that loop formation correlates with cell-type-specific RAD21 signals. **e, f** Peak set enrichment of loop anchor-associated CTCF peaks (equivalent to **c, d**). **g, h** Peak set enrichment of loop anchor-associated H3K27ac regions (equivalent to **c, d**). **I, j** Gene set enrichment of genes with loop anchor-associated TSS (equivalent to **c, d**). **k** Schematic summary of the observed architectural changes during MO differentiation. In general, architectural changes were concordant with changes in gene expression or the activity of regulatory elements, both within domains (loops and TADs), as well as at loop anchors. (**a**, **b**, **d**, **f**, **h**, **j**) Source data are provided as a Source Data file.

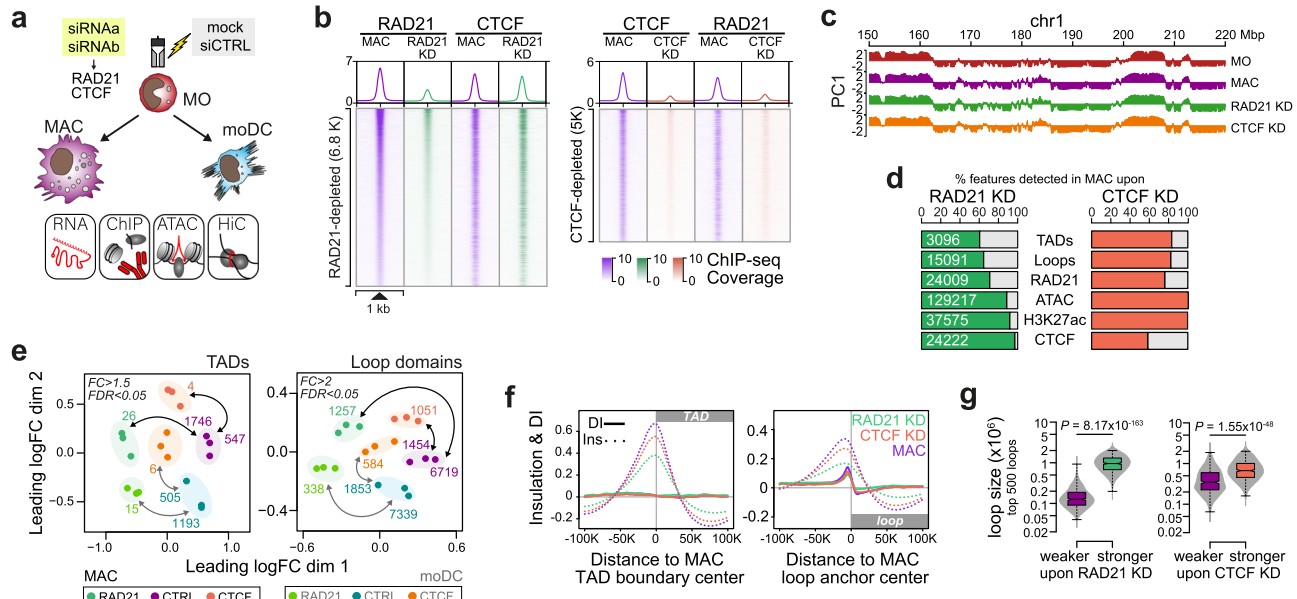

**Fig. 5 Cohesin knockdown affects 3D genome architecture during MO differentiation. a** Schematic of the experimental setup. MO were either mock-treated or electroporated with control siRNA (siCTRL) or specific siRNAs (targeting *CTCF* or *RAD21*) and cultured in MAC or moDC culture conditions. **b** Genomic distance distributions of RAD21 and CTCF ChIP-seq coverage at significantly reduced RAD21 peaks upon *RAD21* knockdown (left panel) or significantly reduced CTCF peaks upon *CTCF* knockdown in MAC. **c** Tracks representing the first eigenvector values (PC1, 50kB resolution) for the indicated cell types. **d** Effect of the *RAD21* or *CTCF* knockdown on indicated features. Colored bars represent the number of detectable features as percentage of the same feature in control MAC. Numbers represent feature counts in control MAC. **e** MDS plots comparing TAD and loop score data sets of the indicated conditions. Numbers of significantly different domains are indicated. **f** Histogram of insulation scores (Ins) and directionality indices (DI) across all loop anchor regions for siRNA-treated or control MAC. **g** Distribution of loop sizes for the top 500 domains gained or lost upon *RAD21* or *CTCF* knockdown in MAC. Solid bars of boxes represent the interquartile ranges (25–75%) with an intersection at the median; whiskers represent max/min values; *P* values: Mann–Whitney U-test, two-sided. (**f**, **g**) Source data are provided as a Source Data file.

both factors was sustained during MO differentiation (RNA levels 7 days after siRNA transfection: MAC *RAD21* 11%, *CTCF* 43%, moDC *RAD21* 18%, *CTCF* 46%). Knockdown of both factors was also reflected by the reduced genome-wide binding of CTCF and RAD21 as measured by ChIP-sequencing (Fig. 5b and Supplementary Fig. 5b). Since we did not have internal controls for normalization, the effects may actually be stronger (normalization tends to attenuate differences in knock-out settings). However, it is clear that even after 7 days of knockdown, both CTCF and RAD21 were still detectable. Binding patterns suggested that some binding sites were more resistant to degradation/turnover than others, a phenomenon that has also been observed in other systems[37,38]. Notably, depletion of RAD21 did not significantly affect the binding of CTCF, while CTCF-depleted sites also lost RAD21 signals (Fig. 5b and Supplementary Fig. 5b), which is in line with the boundary function of CTCF[11,39,40].

To study the effects of *CTCF* and *RAD21* knockdown on genome architecture, we collected additional epigenetic and transcriptome data (via ATAC-, ChIP-, and RNA-sequencing, peak positions and results of differential gene or peak analysis are provided in Supplementary Data File 3, 4) and captured chromatin interactions in both differentiated states (MAC, moDC) using in situ Hi-C (quality metrics are shown in Supplementary Fig. 5c, d).

As already observed in previous studies in other systems[4,41], *CTCF* and *RAD21* knockdown had no significant impact on compartmentalization as measured by the eigenvector values (PC1) (Fig. 5c and Supplementary Fig. 5e). Also, the changes on interchromosomal interactions observed between MO and the two differentiated cell types were not substantially altered in knockdown cells (Supplementary Fig. 5f), suggesting that global rearrangements of chromatin territories that are associated with

nuclear shape-changes proceed either independent of CTCF or RAD21 or early during differentiation before the knockdown made an impact. However, we detected abundant architectural changes on the level of local spatial chromatin organization as summarized in Fig. 5d (Supplementary Fig. 5g for moDC, positions of TADs and loops, as well as RAD21 and CTCF peak positions are provided in Supplementary Data File 4) for feature counts and in Fig. 5e for significantly different TAD and loop scores. We observed a global reduction of TADs and loops, which coincided with a loss of insulation at TAD boundaries and loop anchors, particularly in *RAD21* knockdown cells (Fig. 5f and Supplementary Fig. 5h). When we compared chromatin loops that were lost upon cohesin or CTCF depletion with those that remained (appearing stronger in *RAD21* or *CTCF* KD samples), we noted significant differences in loop sizes. Upon cohesin depletion, cells tended to lose smaller loop domains while strengthened loops appeared larger (Fig. 5g and Supplementary Fig. 5i). To some extend this was also observed in CTCF-depleted cells, in line with the predicted increase of loop sizes upon CTCF depletion[42].

An exemplary interaction map (*LPL* locus on chromosome 8) for MAC and corresponding tracks for epigenome and TF data highlighting these differences is shown in Fig. 6a (for moDC the *CCR1* locus is shown in Supplementary Fig. 6a). Changes in loop anchor strength correlated with RAD21 and CTCF binding data upon knockdown – loop anchors of weakened domains were also enriched for weaker ChIP-seq signals of the corresponding factor and loop anchors in strengthened domains remained stable (Fig. 6b, d and Supplementary Fig. 6b, d). Likewise, the signal strength of RAD21 and CTCF peaks flanking strengthened or weakened loop anchors correlated with their signal in rankings between knockdown and control cells (Fig. 6c, e and Supplementary Fig. 6c, e),

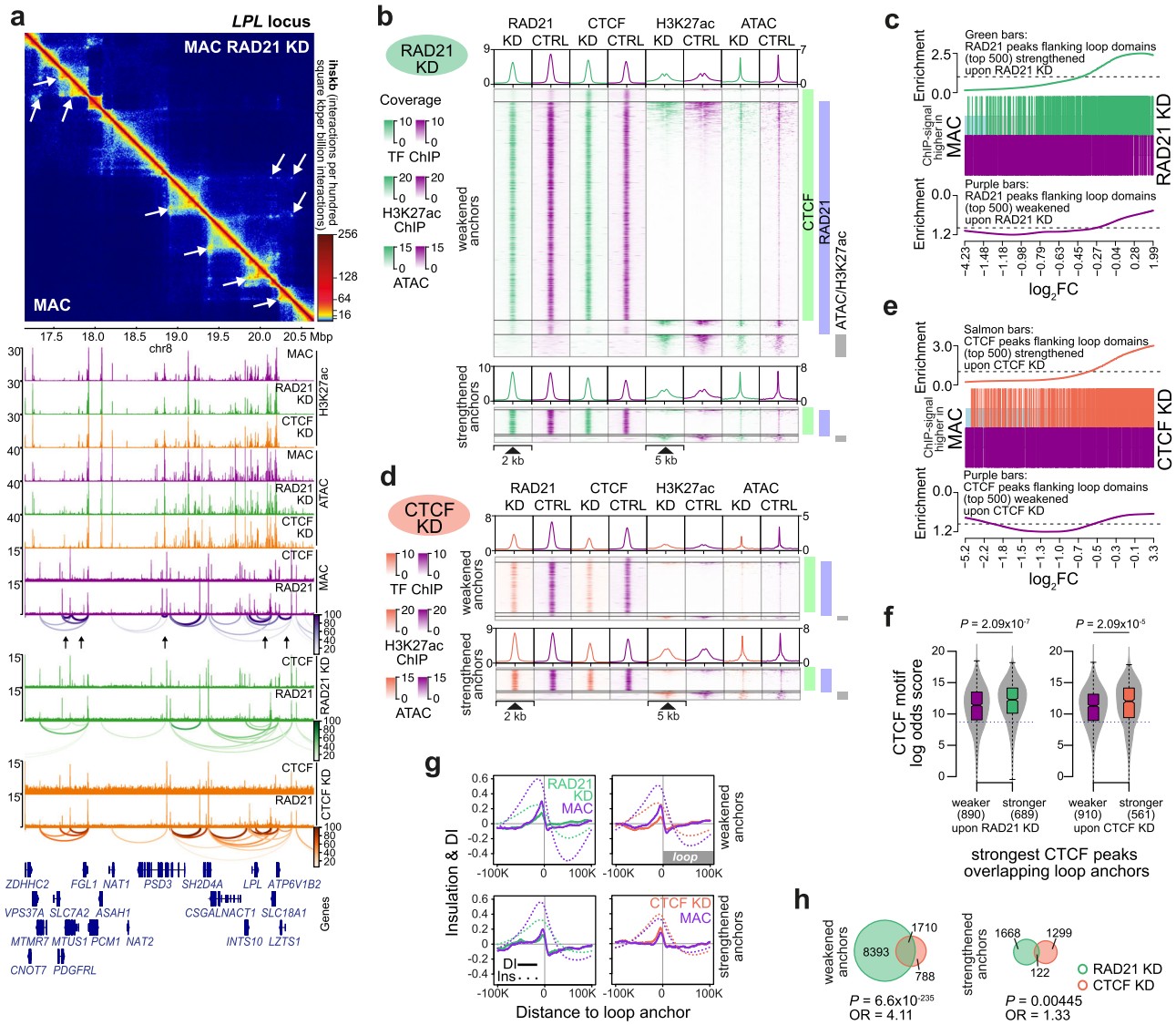

**Fig. 6 Properties of loop domain boundaries altered upon cohesin depletion. a** Comparative in situ Hi-C interaction map and corresponding genome browser tracks for the *LPL* locus on Chr.8. Arrows indicate differential loops. **b, d** Genomic distance distribution across peak-centered loop anchors either strengthened and weakened by *RAD21* (in **b**) or *CTCF* knockdown (in **d**) in MAC, as described in Fig. 4a. **c, e** Peak set enrichment of loop anchor-associated RAD21 (**c**) or CTCF peaks (**e**). Enrichment of loop anchor-overlapping peaks that were altered by *RAD21* or *CTCF* knockdown is plotted across all peaks ranked by their signal in siRNA-treated versus control MAC. **f** Distribution of CTCF motif scores across the indicated peak sets. Solid bars of boxes represent the interquartile ranges (25–75%) with an intersection at the median; whiskers represent max/min values; *P* values: Mann–Whitney U-test, two-sided; dotted line: detection threshold. **g** Histograms of insulation scores and directionality indices across differential loop anchor regions. **h** Venn diagrams, *P* values, and odd ratios for overlaps between loop anchors that are affected by *RAD21* or *CTCF* knockdown in MAC (Fisher's exact test, two-sided). (**b, d, f–h**) Source data are provided as a Source Data file.

suggesting a direct link between the level of CTCF/cohesin occupancy at anchors of loop domains and their strength. The above data clearly suggested different degrees of susceptibility towards CTCF/cohesin loss across loop domains. The comparison between CTCF motif scores (which often correlate with sequence binding affinity) at CTCF binding sites revealed significant differences between anchors of strengthened and weakened loop domains, indicating that differences in binding affinity may contribute to the observed gradient of knock-down susceptibility (Fig. 6f and Supplementary Fig. 6f). In addition, knockdown resistant (strengthened) loops were enriched for CTCF-independent loops. This was particularly true for resistant loops in *CTCF* KDs, where only ~55% of top-ranking anchors were associated with CTCF peaks (compared to ~90% in weakened loops, see peak counts in Fig. 6f and Supplementary Fig. 6f).

The distribution of insulation scores and directionality indices across anchors of differential loops (Fig. 6g and Supplementary Fig. 6g) suggest that the *RAD21* knockdown likely had a much stronger impact than anticipated from the analysis of differential RAD21 peaks or loops, since insulation clearly also dropped at anchors of loop domains that appear strengthened. Hence, while we call differential loops weaker or stronger based on the comparative analysis, in reality (at least for the *RAD21* knockdown) they would be better categorized as less or more resistant, respectively. In a direct comparison of differential loop domains, we observed little overlap between anchors of loops that are strengthened upon *CTCF* or *RAD21* knockdown, while anchors of weakened loops over-lapped substantially (Fig. 6h and Supplementary Fig. 6h), suggesting some level of redundancy, as expected by the pivotal functions of both CTCF and cohesin in defining loop domains.

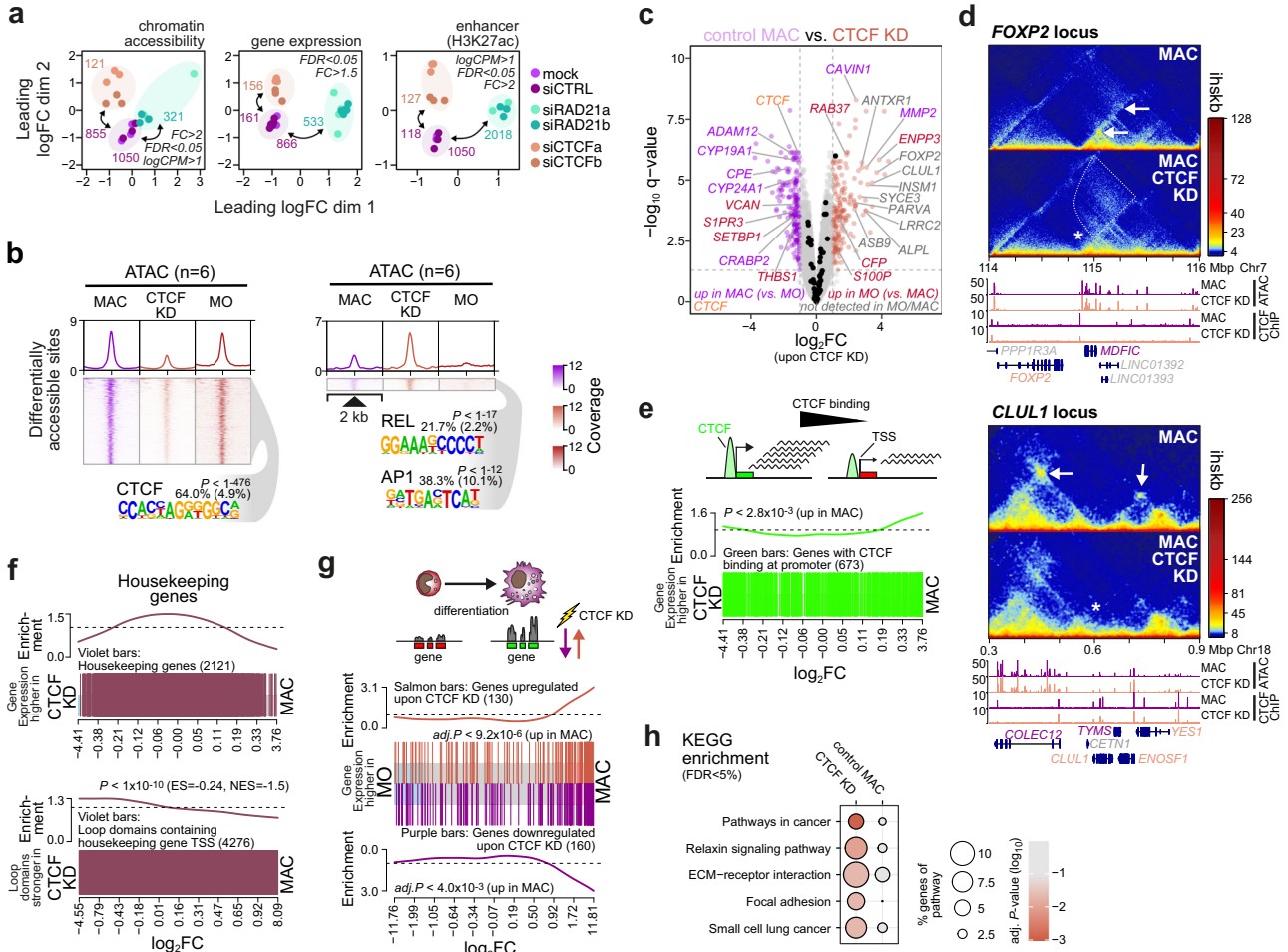

**Fig. 7 CTCF knockdown affects MO differentiation-dependent transcriptional programs. a.** MDS plots of ATAC-seq, RNA-seq and H3K27ac ChIP-Seq data sets for MAC control cells and siRNA knockdowns. Numbers of differentially accessible sites, expressed genes, or H3K27ac marked histones are indicated. **b** Distribution of ATAC-seq signals across differentially accessible sites in control MAC, *CTCF* siRNA-treated cells as well as freshly isolated MO. *De novo*-derived motifs for each cluster are given along with the significance of motif enrichment (hypergeometric test) and the fraction of motifs in peaks (background values are in parenthesis). **c** Volcano plot of genes differentially expressed in control versus *CTCF* knockdown MAC. Exemplary genes are highlighted. Purple dots, downregulated upon KD. Salmon dots, upregulated upon KD. Black dots, top 50 upregulated genes during normal MO differentiation. Gene symbols in gray, not detected in MO or control MAC. Gene symbols in red, genes downregulated during MO differentiation. Gene symbols in purple, genes upregulated during MO differentiation. **d** Comparative in situ Hi-C interaction maps and corresponding genome browser tracks for example regions of dysregulated genes. Stars indicate CTCF-bound boundaries, arrows indicate differential loops and the dotted line marks contacts that were induced upon *CTCF* KD. **e** Enrichment of genes with promoter-associated CTCF peaks across all genes ranked by their signal in *CTCF* siRNA-treated versus control MAC (P value: two-sided rotation test). **f** Analysis of housekeeping genes. Top panel: Gene set enrichment across all genes ranked by their signal in siRNA-treated versus control MAC (no significant enrichment using two-sided rotation tests). Bottom panel: loop set enrichment for domains containing housekeeping genes across all loop domains ranked by their score in siRNA-treated versus control MAC (ES, enrichment score; NES, normalized enrichment score; P determined by permutation test). **g** Gene set enrichment of knockdown regulated genes during MO differentiation (P-values: two-sided rotation tests, BH correction). **h** KEGG pathway analysis of genes affected by CTCF loss in MAC (P-values: hypergeometric tests, BH correction). (**b**, **e**, **f**, **h**) Source data are provided as a Source Data file.

These analyzes showed that knockdown of *RAD21* and *CTCF* clearly affected chromatin organization during MO differentiation on the levels of TADs and loops, although at different degrees. 3D genome architecture in MO-derived MAC and moDC is most affected upon cohesin depletion with loops on sub-TAD levels losing connectivity.

**Effects of *CTCF* knockdown on transcriptional landscapes.** We next asked, whether these architectural changes would also affect transcriptional programs. Figure 7a illustrates differences in chromatin accessibility, gene expression and H3K27ac deposition between control MAC and knockdown cells, which were

abundant and significant (corresponding data for moDC are shown in Supplementary Fig. 7a). Super-enhancers were only marginally affected (no significantly different SE in comparisons of *CTCF* KD samples and only few in *RAD21* KD). Effects of the *CTCF* knockdown on MO differentiation were generally more moderate compared to the cohesin knockdown. As indicated by the de novo motif analysis shown in Fig. 7b (Supplementary Fig. 7b for moDC), chromatin accessibility changes in the *CTCF* knockdown were dominated by CTCF sites, suggesting that the activities of other transcription factors driving MO differentiation were not drastically altered. Accessible sites (as well as H3K27ac regions) lost upon *CTCF* knockdown, were significantly enriched at loop anchors, but motif composition was not different between

loop anchor-associated and non-associated sites (data shown in Supplementary Tables 3 and 4).

Gene expression profiles indicated the aberrant expression of "non-lineage" genes in MAC upon CTCF-depletion. Prominent examples are marked in gray in the volcano plot in Fig. 7c (corresponding data for moDC are shown in Supplementary Fig. 7c). The observed dysregulation is most likely explained by the weakening of CTCF boundaries, which increases aberrant contacts of these "non-lineage" genes (examples are shown in Fig. 7d). It is known that CTCF can have boundary-independent functions at gene promoters[43,44], hence we tested how genes with CTCF binding at their proximal promoters would be affected by CTCF knockdown. As shown in Fig. 7e (and Supplementary Fig. 7d for moDC), these genes tended to be downregulated upon CTCF knockdown, suggesting that boundary-independent functions of CTCF contribute to the altered transcriptional profiles observed after CTCF knockdown. As shown Fig. 7f (Supplementary Fig. 7e for moDC), the expression of housekeeping genes (as defined by the HRT Atlas[45]) was least affected by the CTCF knockdown and loop domains containing them tended to remain stable upon knockdown. Interestingly, depletion of CTCF mainly affected genes that were normally induced during MO differentiation (Fig. 7g and Supplementary Fig. 7f for moDC). Pathways were only significantly enriched in genes upregulated during knockdown, likely reflecting the aberrant gene induction in these cells (Fig. 7h and Supplementary Fig. 7g for moDC). The complex relationships between CTCF knockdown and associated changes in chromatin interaction profiles and transcription suggest locus-specific contributions of CTCF both as insulator or transcription factor, which may entail divergent effects on transcription upon CTCF knockdown. While the effects of CTCF knockdown were moderate, they were clearly significant. It is likely that changes would increase with a more rapid and complete CTCF depletion, which has so far not been feasible in human primary MO.

**Effects of cohesin knockdown on transcriptional landscapes.** Notably, the composition of differential accessible chromatin regions was different in RAD21 and CTCF knockdown MAC. Cohesin depletion clearly affected sites that normally gain accessibility during MO differentiation into MAC (Fig. 8a). Accessibility changes were less pronounced in moDC, which may relate to quality issues in moDC ATAC-seq data (Supplementary Fig. 8a, for QC data see Supplementary Table 10). In regions that gained accessibility, we observed an enrichment of NFκB (REL) motifs (Fig. 8a), while regions that were lost upon RAD21 knockdown (as shown in Fig. 8a, b) resembled the MAC-specific signature observed during normal MO differentiation (Supplementary Fig. 1e), except for CTCF. The latter was particularly enriched in loop anchor-associated, accessible regions that were lost upon RAD21 knockdown (compared to corresponding regions not associated with loops, see Supplementary Table 3). Contrasting the CTCF knockdown results, in which CTCF-dependent accessible regions were enriched in loop anchors, cohesin-dependent accessible regions were under-represented at loop anchors (data shown in Supplementary Table 3). Hence, accessibility changes during RAD21 knockdown occur mainly in regulatory elements that were not directly involved in loops that are detected with in situ Hi-C.

During normal differentiation, we had observed several regions for which the appearance of de novo loops coincided with accessibility changes (see Supplementary Fig. 3h), suggesting the creation of novel cohesin-loading sites. Interestingly, while these loops were not generated (as expected) during RAD21 knockdown, differentiation-associated chromatin accessibility changes were still observed (Fig. 8c).

In line with the altered motif signatures in differentially accessible chromatin, RAD21 knockdown in MAC clearly affected gene expression programs during MO differentiation (as indicated by purple and red gene symbols in the volcano plot shown in Fig. 8d). Genes that were downregulated in MAC upon RAD21 knockdown were strongly enriched for genes upregulated during MO differentiation (purple bars and curve in Fig. 8e), while upregulated genes in MAC upon RAD21 knockdown were enriched for genes generally regulated during differentiation (both up and down, green bars and curve in Fig. 8e) as well as some "non-lineage" genes (gray gene symbols in Fig. 8d). Concordantly, the expression of housekeeping genes was least affected by the RAD21 knockdown and loop domains containing them tended to remain stable upon knockdown (Fig. 8f). In line with this, RAD21-strengthened loops tended to be enriched for unaffected genes, while weakened loops tended to be enriched for RAD21 sensitive genes (Fig. 8g) and RAD21 sensitive TADs were significantly enriched for genes downregulated upon RAD21 knockdown (Fig. 8h). Pathway enrichment identified several MAC-relevant pathways (e.g., ECM-receptor interaction, protein digestion or steroid biosynthesis) as being enriched genes affected in RAD21 knockdown (Fig. 8i). Similar observations were also made in moDC (see Supplementary Fig. 8b–g). Hence, the observed transcriptional changes suggested that cohesin depletion significantly affected differentiation-associated regulation.

Changes in transcription programs in MAC were associated with corresponding changes in the activity of gene regulatory elements (as detected by H3K27ac ChIP-seq). Regions with decreased H3K27ac upon RAD21 knockdown showed a significant positive correlation with increased H3K27ac during MO differentiation (Fig. 8j, purple bars and curve), while regions with increasing H3K27ac upon RAD21 knockdown showed a significant correlation with decreasing H3K27ac during MO differentiation (Fig. 8j, green bars and curve), further supporting the requirement for cohesin for the normal MO differentiation program (results for moDC are shown in Supplementary Fig. 8h). To find out whether these changes in the enhancer landscapes were associated with characteristic transcription factor signatures, we performed de novo and known motif searches across accessible regions in differential H3K27ac regions. Here, we split regions based on their activity in MO. De novo motif signatures were partially overlapping between weakened and strengthened loops, with ETS (PU.1), CEBP and AP1 motifs being enriched (over genomic background) in all types of differential enhancers (Fig. 8k). In direct comparisons (Fig. 8l), we observed the enrichment of NFκB (REL) and AP1 motifs in strengthened enhancers (in line with signatures observed in differentially accessible sites), while weakened enhancers were enriched in CTCF motifs (Fig. 8l). The latter is in line with a significant overlap of weakened H3K27ac regions with loop anchors (data shown in Supplementary Table 4).

Similar trends were observed in moDC (Supplementary Fig. 8a–j), except for motif signatures. At enhancers strengthened in moDC upon RAD21 knockdown, we observed the enrichment of ETS:IRF (EIRE) and EGR motifs, while weakened enhancers were enriched in CTCF, KLF and AP1 motifs (Supplementary Fig. 8j). Previous work in mouse macrophages demonstrated distinct transcriptional functions of AP1 family members[46]. The difference in the distribution of AP1 motifs in strengthened and weakened anchors between MAC and moDC likely points to differing activities of individual AP1 family members in these cell types.

Taken together, our analyzes of genome organization, as well as transcriptional and enhancer landscapes showed that the reduction of cohesin has profound effects on genome structure and transcription programs associated with differentiation.

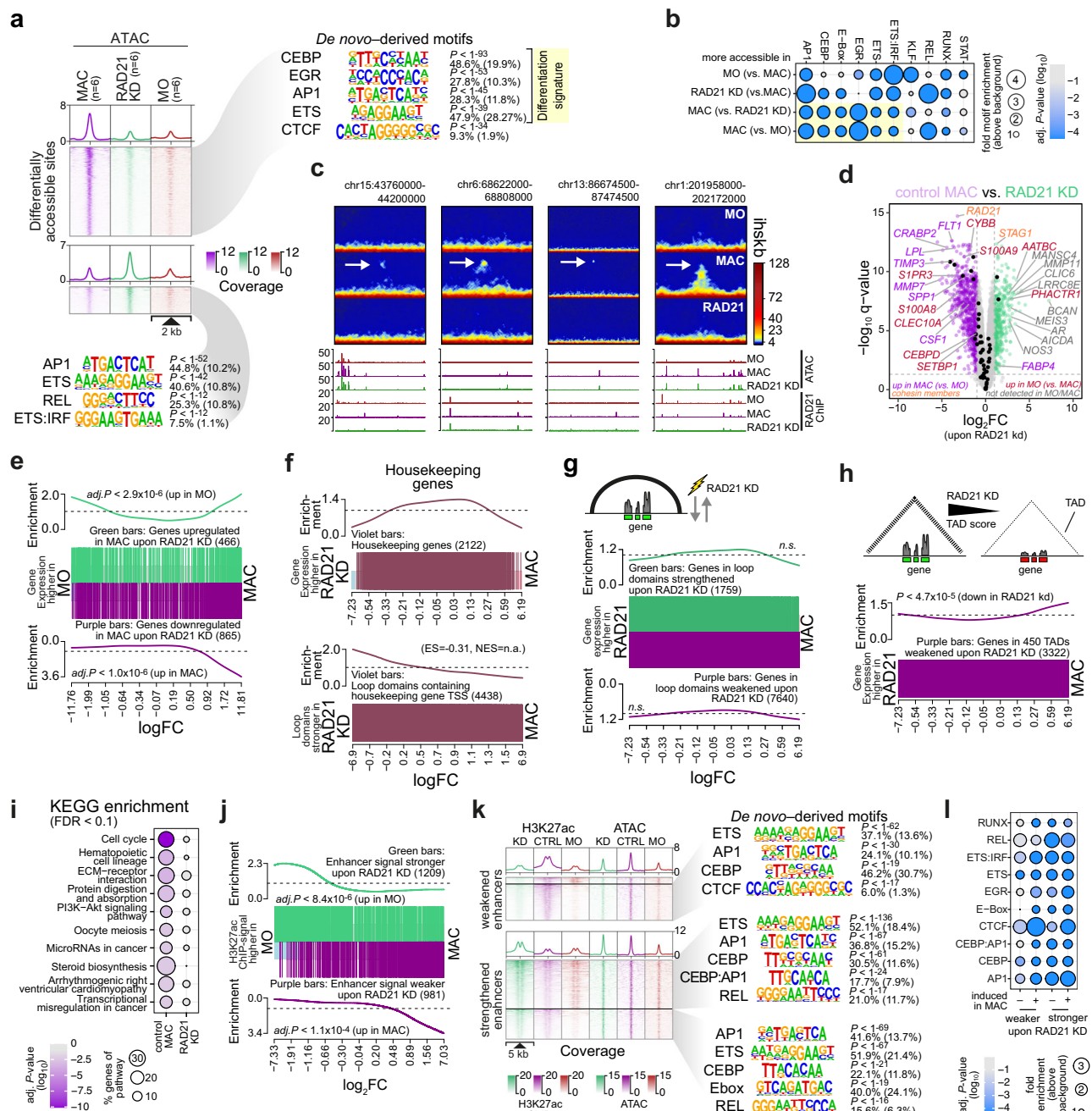

**Fig. 8 Loss of cohesin profoundly affects MO differentiation-dependent transcriptional programs. a** Distribution of ATAC-seq signals across differentially accessible sites in control MAC, *RAD21* siRNA-treated cells as well as freshly isolated MO. Top *de novo*-derived motifs for each cluster are given (as in Fig. 7b). Signature motifs that are associated with MO differentiation are highlighted. **b** Comparative enrichment of known TF motifs across differentially accessible chromatin regions of the indicated comparisons. **c** Comparative in situ Hi-C interaction maps and corresponding genome browser tracks for example regions of differentiation-associated loops. Arrows mark differential loops that were induced upon MO differentiation. **d** Volcano plot of genes differentially expressed in control versus *RAD21* knockdown MAC, as described in Fig. 7c. **e** Enrichment of genes that were differentially expressed upon *RAD21* KD across all genes ranked by their signal in MO versus MAC. **f** Analysis of housekeeping genes. Top panel: Gene set enrichment of all genes ranked by their signal in siRNA-treated versus control MAC. Bottom panel: loop set enrichment for domains containing housekeeping genes across all loop domains ranked by their score in siRNA-treated versus control MAC (ES, enrichment score; NES, normalized enrichment score; *P* not determined due to unbalanced gene-level statistics). **g** Enrichment of genes in altered loop domains across all genes ranked by their signal in siRNA-treated versus control MAC. **h** Gene set enrichment of genes in weakened TADs across all genes ranked by their signal in siRNA-treated versus control MAC. **i** KEGG pathway analysis of genes affected by RAD21 loss in moDC. **j** Enrichment of H3K27ac-marked regions affected by *RAD21* KD across all enhancers ranked by their signal intensity in MO versus MAC. **k** Genomic distance distributions of H3K27ac and ATAC signals at differential H3K27ac peaks, centered on overlapping open chromatin. Top *de novo*-derived motifs are presented as in (**a**). **l** Motif enrichment across open chromatin associated with H3K27ac peak regions as indicated. *P*-values were determined using hypergeometric tests in (**a**, **b**, **i**, **k**, **l**) or two-sided rotation tests in (**e**, **g**, **h**, **j**) and adjusted for multiple testing (BH correction) except in (**a**, **h**, **k**). (**a**, **b**, **i**, **k**, **l**) Source data are provided as a Source Data file.

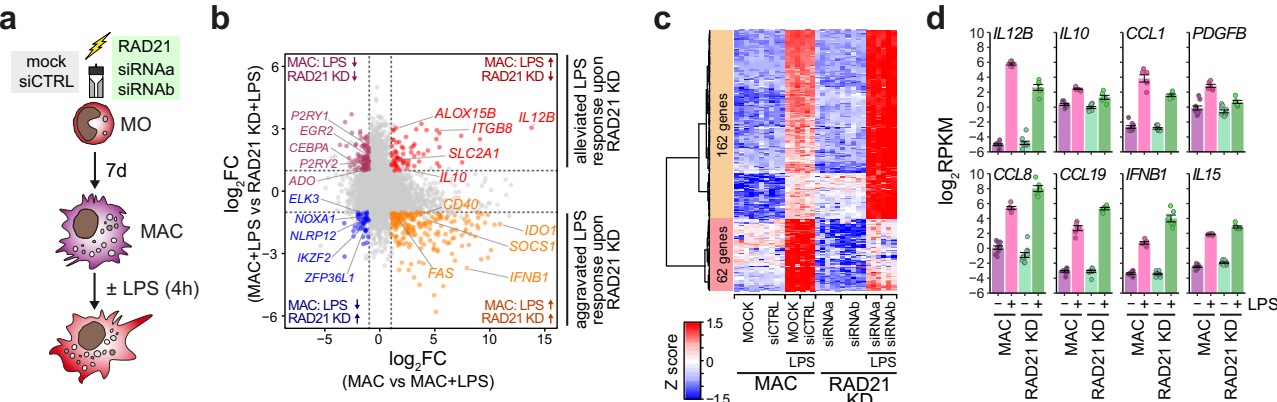

**Fig. 9 Loss of cohesin alters innate activation programs in MAC. a** Schematic of the experimental setup. MO were either mock-treated or electroporated with control siRNA (siCTRL) or either of the two specific *RAD21* targeting siRNAs and cultured in MAC culture conditions. MAC were activated with LPS for 4 h on day 7. **a** Scatter plot of genes differentially expressed in control versus *RAD21* knockdown MAC upon LPS stimulation. Genes were categorized for their response to LPS in control MAC (x-axis) and for the effect of *RAD21* KD after LPS stimulation (y-axis). Colored genes showed at least two-fold difference in their expression (FDR < 0.05) in both comparisons, as indicated in the corners of the plot. Exemplary genes are highlighted. **c** Heatmap presenting scaled expression data of differentially LPS-induced genes (FC > 2, CPM > 1, FDR < 0.05, quasi-likelihood (QL) F-test for 1.5-fold difference) between control- (MOCK & siCTRL) and *RAD21* KD cells. Independent replicates per condition: unstimulated cells, n = 5; LPS-treated cells, n = 3. **d** Bar plots showing expression levels examples of genes differentially activated upon *RAD21* KD (mean normalized RPKM ± SE based on the RNA-seq data; controls: n = 10 samples from five donors; LPS-treated: n = 6 samples from three donors). All examples were significantly induced upon LPS treatment in MAC (FDR < 0,05, QLF-test for 1.5-fold difference) and showed significant differences in their expression between LPS-stimulated control and *RAD21* KD MAC (FDR < 0,05, QLF-test for 1.5-fold difference). (**b–d**) Source data are provided as a Source Data file.

Cohesin depletion strongly reduces contact frequencies and the insulation of contact domains which leads to the dysregulation of enhancers and genes that are significantly associated with the normal differentiation process.

**Innate activation programs are altered in cohesin-depleted MAC.** A hallmark function of MO and MAC is their ability to induce innate inflammatory responses upon pathogen encounter, which are primarily regulated through NFκB-dependent pathways. To investigate whether cohesin depletion during the differentiation of primary human MO would affect the functionality of differentiated MO-derived MAC, we compared the immediate response of control MAC with *RAD21* knockdown MAC, both exposed to bacterial lipopolysacharide (LPS) for 4 h (Fig. 9a). Control MAC and of *RAD21* knockdown MAC induced overlapping transcriptional programs (as demonstrated by GSEA in Supplementary Fig. 9a, b), which were, however, quantitatively different. The scatter plot in Fig. 9b depicts LPS-induced changes in control MAC contingent on the differences observed between LPS-stimulated MAC and cohesin-depleted MAC (*RAD21* KD). As also seen in the heatmap in Fig. 9c, some genes were consistently less induced in cohesin-depleted MAC, including *IL12B* and *IL10* (examples are given in Fig. 9d, top panel), while the larger fraction of induced genes was even more induced in cohesin-depleted MAC (Fig. 9d, bottom panel), including *IFNB1*. Other prominent inflammatory cytokine genes like *IL6* and *TNF*, however, were not significantly altered (Supplementary Fig. 9c). This contrasts with previous observations in mouse bone-marrow derived macrophages (BMM), where the *Rad21-/-* BMM were generally less responsive towards LPS stimulation[16]. To allow the side-by-side comparison of altered LPS-responses between macrophages of both species, we re-analyzed the published mouse data and compared gene expression data of both species across orthologous genes. Corresponding heatmaps (Supplementary Fig. 9d, e; either based on the differential human gene set shown in Fig. 9c, or on the original clustering shown in Fig. 2 of Cuartero et al[16].) as well as gene expression profiles of individual target genes (Supplementary Fig. 9f) highlight the different effects

of cohesin loss on LPS response between both species. Hence, innate inflammatory responses are not blunted in cohesin-depleted human MO-derived MAC, as described for mouse BMM[16], but significantly altered and even partly enhanced.

## Discussion

This study provides a comprehensive analysis of cohesin and CTCF functions in differentiating MO. We show that normal postmitotic differentiation of MO entails significant alterations in nuclear organization, including widespread changes in chromatin looping. Both acquired and lost loops were enriched in genes and enhancers that were also regulated during MO differentiation, suggesting that conformational changes are linked with transcriptional changes during MO differentiation. To further evaluate their relationship, we depleted the CCCTC-binding factor CTCF and the cohesin complex component RAD21 during MO differentiation. While the depletion of CTCF had a minor impact on enhancer and transcription landscapes, the knockdown of *RAD21* had a more pronounced impact: cohesin depletion resulted in loop dispersion, as well as an altered enhancer and gene activation, which was associated with altered transcription factor signatures. Strikingly, dysregulated genes and enhancers were strongly enriched for those regulated during differentiation. In addition, innate activation programs were altered in cohesin-depleted MO-derived MAC, suggesting that cohesin is required for functional MO differentiation and activation.

The changes during human MO differentiation resemble those observed in other developmental[33,47–50] or trans-differentiation[51] models, where architectural changes generally coincided with transcriptional changes. Studies in various differentiation models, including ESC-derived cells[34], or T cells[49] had identified specific transcription factors that were enriched at cell stage-specific loop anchors. In a leukemia model of monocyte-macrophage differentiation (PMA-treatment of the THP-1 cell line) for example, it was shown that dynamic looping events were preferentially connecting regulatory elements that were enriched in AP1 binding sites in the differentiated state[50]. In our primary cells, CTCF-independent loops were generally enriched for motifs

corresponding to TFs that were also found in cell-type-specific enhancer regions (including PU.1, AP1, etc.), suggesting that it is not a single factor that drives the formation of dynamic regulatory loops. This is also in line with promoter-capture Hi-C data generated from 17 blood cell types, which found both H3K27ac and H3K4me1 enriched at promoter-interacting sites[52]. In contrast to the THP-1 model, loops that were gained during post-mitotic differentiation into MAC or moDC were to the large extent CTCF-anchored. Some of the observed differences may be attributed to the very high Hi-C sequencing coverage in the published THP-1 study (>five billion reads per cell state), which likely improved the detection of regulatory loops. However, the predominance of structural loops in macrophages may point to biological rather than technical differences between the primary and the cell line model of MAC differentiation. Notably, MO are already postmitotic and functional effector cells and their differentiation covers a much smaller developmental scale compared to stem cell-based models. This is mirrored by the lack of marked A/B compartment switches, which are more common in stem cell differentiation models[34,53]. Interestingly, we observed the appearance of contact loops in regions with low or no transcriptional activity. Many of these de novo loops disappeared upon cohesin-depletion and were characterized by the co-appearance of chromatin accessibility. This suggests that the newly remodeled sites may serve to load cohesin before it extrudes the novel chromatin loops. Remodeling has indeed been functionally associated with cohesin. Mutations in cohesin components or several subunits of the SWI/SNF chromatin-remodeling complex are associated with neurodevelopmental disorders in humans that present overlapping clinical phenotypes[54]. More recent work in neutrophils showed that both ATP-dependent chromatin remodelers BRG1 and BRM were required for the Calcium-induced recruitment of the cohesin loading factor NIPBL to promoter-distal elements[55]. Given that the majority of chromatin loops acquired during MO differentiation were structural and confined by a pair of convergent CTCF sites, it is conceivable that they were created via novel loading sites that became available during MO differentiation. In such a scenario, cohesin loading would follow the action of transcription factors that recruit remodeling complexes and other cofactors like NIPBL to rewire loop architecture during differentiation.

The observed mild phenotype of CTCF-depletion during MO differentiation is in line with a recent study demonstrating minor transcriptional effects of Auxin-mediated CTCF depletion during CEBPA-induced transdifferentiation of B-cells into macrophages[56], suggesting that CTCF (and its function as an insulator) may be dispensable for MO differentiation. However, given the incomplete removal of CTCF via the siRNA approach and the presence of (at least partially) knockdown-resistant CTCF sites[38], we cannot rule out that complete depletion of CTCF (although difficult to achieve in primary MO) may have a stronger impact in the primary differentiation model. It should generally be noted that the knockdown approach used in the present study entails the typical siRNA-inherent limitations. While we can effectively reduce mRNA levels, we are unable to control the availability of the targeted proteins, which is entirely dependent on their degradation rate. Hence, the effects of knockdown will gradually increase with protein turnover and will be less evident early after culture, where protein levels for CTCF and RAD21 are less affected. In addition, we observe a mixture of cohesin or CTCF sites that are either sensitive or (at least partially) resistant to knockdown, a phenomenon that has also been observed in earlier studies in other systems[37,38]. Consequently, early events during MO differentiation likely proceed normally. This may include the early changes in nuclear shape (and the

corresponding territorial changes), which were not different between control and knockdown cells, as well as early changes in cohesin-mediated looping. Given the relatively long culture (7 days) it is also likely that some of the effects observed on the level of chromatin accessibility or transcription regulation are secondary. A further limitation in our study is the lack of internal controls (e.g. RNA spike-in controls, or barcoded control histones for H3K27ac ChIP assay) which could have improved data normalization. Despite these limitations, this work clearly shows that sufficient cohesin levels are required for normal MO differentiation. As previously noted[13], the observed sensitivity towards cohesin loss was limited to a subset of genes expressed in MO and MO-derived cells. Our analyzes further suggest that genes affected by cohesin depletion are enriched for dynamically regulated genes and concurrently depleted for housekeeping genes, which is in line with the observation that housekeeping genes are generally less connected to distal enhancers compared to regulated, dosage-dependent genes[15].

Along the same lines, we also show that cohesin-depleted MAC display a strongly altered innate immune activation profile. Here, genes induced by the inflammatory stimulus may be affected on various levels: 1) Their response may be affected by the altered chromatin conformation and interaction landscapes in MAC that were differentiated in the presence or absence of cohesin. 2) Their response may also be altered due to secondary alterations in signaling networks required to maintain the normal response. 3) Their response may be directly affected by cohesin depletion, which may prevent the formation of LPS-induced loops that are necessary for a normal response. Interaction landscapes can rapidly change during macrophage stimulation, as exemplified by conformation changes observed after 2 h of IFNγ stimulation in mouse macrophages[57] or 2 h of LPS treatment of THP1 cells[58]. NFκB and AP-1, the main TF driving the early responses to pathogen-associated molecular patterns like LPS, frequently bind promoter-distal sites[59] and may require cohesin to properly interact with their target promoters during activation. Interestingly, in our human post-proliferative MO differentiation model, the changes include both the up- and down-regulation of inflammatory response genes. This is in contrast with previous observations in mouse MAC[16], where cohesin depletion generally blunted inflammatory gene activation. One possible explanation for this difference may be the higher level of *IFNB1* expression in cohesin-depleted human MAC, while the counterpart in mice appeared deficient in IFN signals and IFNβ treatment was able to partially restore the inflammatory response in cohesin-depleted mouse MAC[16]. Some of the differences may also be attributed to species divergent LPS responses, as previously described[60].

In conclusion, we demonstrate that the siRNA-mediated depletion of cohesin interferes with normal human MO differentiation as well as innate immune functions of human MO-derived macrophages. The expression of housekeeping genes appears to be less affected by cohesin depletion, while the expression of dynamically regulated genes (differentiation-associated or activation-induced) was more dependent on cohesin. Our work also implies that MO and their progeny derived from cohesin-mutated myeloid precursors may have defects both in differentiation and activation programs, which may contribute to disease pathologies.

## Methods

**Cells**. Collection of blood cells from healthy donors was performed in compliance with the Helsinki Declaration. All donors signed an informed consent. Blood sampling, the leukapheresis procedure, and subsequent purification of peripheral blood monocytes were approved by the ethical committee of the University of Regensburg (reference number 12-101-0260). Donors received an expense allowance, as approved by the ethical committee. Blood MO were separated by leukapheresis of male healthy donors followed by density gradient centrifugation over

Ficoll/Hypaque and subsequent countercurrent centrifugal elutriation. MO was either mock electroporated or electroporated with a control siRNA (siCTRL) before culture. As shown previously[27], electroporation does not have a major impact on cell survival or differentiation of human MO. For in vitro differentiation of MO-derived dendritic cells (moDC), MO were cultured in RPMI 1640 (Thermo Fisher Scientific) supplemented with 280 U/ml GM-CSF (Berlex, Seattle, USA) and 20 U/ml IL-4 (Promokine)) and 10% fetal calf serum (Sigma-Aldrich, Taufkirchen, Germany). For in vitro differentiation of MO-derived macrophages (MAC), MO were cultured in RPMI 1640 (Thermo Fisher Scientific) supplemented with 2% human AB serum (Bavarian Red Cross). For LPS stimulation, $2 \times 10^6$ mock electroporated or siRNA transfected MO were differentiated into MAC in tissue-culture treated plates. On day 7 of culture, Lipopolysaccharide (LPS, *Salmonella abortus equi*, Enzo Life Sciences) was added at a concentration of 100 ng/μL for 4 h. Cell viability was monitored using trypan blue staining after harvesting.

**Transient transfections**. siRNA-mediated knockdown experiments were performed as previously described[27] with slight modifications. In brief, transient transfections were performed using chemically modified siRNAs (Axolabs) against *CTCF* (si3444 and si3754), *RAD21* (si467 and si2031), and luciferase (control siRNA). siRNAs-sequences are listed in Supplementary Table 5. All siRNA transfections were performed immediately after MO isolation prior to cell-type-specific differentiation. Cells treated with siRNAs were harvested at day 7 of differentiation. Prior to electroporation, cells were washed once with phenol red-free RPMI 1640 (Gibco) and once with phenol red-free Opti-MEM I (Gibco) at room temperature. After centrifugation, cells were gently resuspended in phenol red-free Opti-MEM I at a final concentration of $10 \times 10^6$ cells in a final volume of 200 μl. For electroporation in a 4 mm cuvette, 200 μl of resuspended cells were combined with 10 μg of the respective siRNA. Electroporation was performed using a Gene Pulser Xcell (BioRad) with a rectangular pulse of 400 V and 5 ms duration. Transfected cells were immediately placed in shake flasks containing pre-warmed culture medium at a density of $1 \times 106/ml$ and cultured at 37 °C, 5% CO2, 95% humidity for 7 days.

**In Situ Hi-C**. In situ Hi-C was essentially performed as described[61] with slight modifications. For robustness, three biological replicates of three independent donors were used for each condition. Cells were pelleted by centrifugation and crosslinked in 1% formaldehyde in PBS for 10 min at room temperature. The reaction was quenched by adding 1/20th volume of 2.625 M glycine, and fixed cells were washed twice with ice-cold 0.5% BSA in PBS. Aliquots of 2 M cells each were snap-frozen and stored at −80 °C. After thawing cell pellets on ice, nuclei were isolated by resuspending the cell pellet in 200 μl wash buffer 1 (50 mM Tris/HCl pH 7.5, 10 mM NaCl, 1 mM EDTA, 0.5% SDS, 1x protease inhibitor cocktail (Roche)). Nuclei were incubated at 62 °C for 7 min in a PCR cycler and spun down at $2500 \times g$ for 5 min at room temperature. Most of the supernatant was discarded, leaving the nuclei in 10 μl of liquid. Samples were resuspended in 245 μl reaction buffer (25 μl 10% Triton X-100, 25 μl Dpn II buffer (NEB), 195 μl water) and rotated at 37 °C for 15 min.

Chromatin was digested overnight with 2 μl (100 U) Dpn II (NEB) at 37 °C in a thermocycler (Eppendorf). The next day, nuclei were collected by centrifugation and 225 μl of the supernatant was discarded, leaving the nuclei in 25 μl liquid. Overhangs were filled in with Biotin-14-dATP (Invitrogen) by adding 75 μl Klenow Master Mix (54.45 μl water, 7.5 μl NEBuffer 2, 1.05 μl 10 mM each dCTP/dTTP/dGTP, 7.5 μl 0.4 mM Biotin-14-dATP, 2 μl (10 U) Triton X-100, 2.5 μl (12.5 U) Klenow fragment (Enzymatics) and rotated overhead at room temperature, 8 rpm for 40 min. Reactions were stopped by adding 2.5 μl 0.5 M EDTA and placed on ice. Proximity ligation was performed by transferring the entire reaction into 1.5 ml Eppendorf tubes containing 400 μl ligase mix (322.5 μl water, 40 μl 10x T4 DNA ligase buffer (Enzymatics), 36 μl 10% Triton X-100, 0.5 μl 10% BSA, 1 μl (600 U) T4 DNA ligase (HC, Enzymatics) and incubating overnight at 16 °C in a thermocycler. Reactions were stopped with 20 μl 0.5 M EDTA containing 1 μl 10 mg/ml DNase-free RNase A for 15 min at 42 °C. Then 33 μl 5 M NaCl and 55 μl 10% SDS were added and crosslinks were reversed for 4 h at 65 °C. Proteins were digested with 10 μl 20 mg/ml DNase-free proteinase K (Roche) for 2 h at 55 °C shaking at 850 rpm, followed by 90 min at 65 °C shaking at 850 rpm. After extraction with 600 μl pH 8-buffered phenol/chloroform/isoamyl alcohol (Thermo Fisher) followed by 300 μl chloroform, DNA was precipitated with 1.5 μl (22.5 μg) Glycoblue (Thermo Fisher) and 1400 μl 100% ice-cold ethanol overnight at −20 °C, pelleted for 20 min at $13000 \times g$, 4 °C, washed twice with 80% ice-cold ethanol, and the pellet was air-dried and dissolved in 131 μl TT (0.05% Tween 20, 10 mM Tris pH 8). DNA concentration was analyzed using the Qubit HS DNA Kit (Thermo Fisher Scientific) and 800 ng of DNA in 130 μl TT was sheared to 300 bp on a S2 focused-ultrasonicator (Covaris) for 80 seconds according to manufacturer's instructions. Sheared-DNA was bound to 20 μl pre-washed Streptavidin-coated Dynabeads (MyOne Streptavidin T1, Thermo Fisher Scientific). The binding reaction was rotated for 30 min at room temperature, and DNA-bound beads were washed once with 400 μl 1x B&W buffer (2x B&W buffer: 10 mM Tris/HCl, pH 7.5, 1 mM EDTA, 2 M NaCl) containing 0.1% Triton X-100 and once with TET (0.2% Tween 20, 10 mM Tris pH 8, 1 mM EDTA). Beads were resuspended in 100 μl Blunting Mix (81.1 μl water, 0.5 μl 10% Tween 20, 10 μl 10x T4 DNA ligase buffer (Enzymatics), 4 μl 10 mM dNTP, 2 μl (6 U) T4 DNA polymerase (Enzymatics),

0.4 μl (2 U) Klenow fragment (Enzymatics), 2 μl (20 U) T4 polynucleotide kinase (Enzymatics)) and incubated for 30 min at 20 °C in a PCR cycler. Reactions were stopped by adding 2.5 μl 0.5 M EDTA, beads were collected on a magnet and washed twice with 150 μl 1x B&W containing 0.1% Triton X-100, and once with 180 μl TET. Beads were resuspended in 50 μl A-tail Mix (41.3 μl water, 0.5 μl 10% Tween 20, 5 μl NEBuffer 2, 0.2 μl 100 mM dATP, 3 μl (15 U) Exo-Klenow (Enzymatics) and incubated for 30 min at 37 °C in a PCR cycler. Reactions were stopped by adding 1.5 μl 0.5 M EDTA, beads collected on a magnet and the supernatant was discarded. Beads were resuspended on ice in 48 μl ice-cold Ligation mix (22.5 μl water, 25 μl 2x T4 DNA ligation buffer (Enzymatics), 0.5 μl 10% Tween 20), and 0.8 μl 25 μM Bioo Nextflex DNA sequencing adapters were added prior to adding 1 μl (600 U) T4 DNA Ligase (HC, Enzymatics), and the reaction incubated for 20 min at room temperature. Reactions were stopped by adding 5 μl 0.5 M EDTA, and beads were washed twice with 1x B&W containing 0.1% Triton X-100, once with TET and resuspended in 30 μl LoTET (TET diluted 1:4 with water). Libraries were amplified by PCR for 12 cycles (98 °C, 30 s; 12x [98 °C, 10 s; 60 °C, 25 s; 72 °C, 30 s]; 72 °C, 5 min; 4 °C, ∞ ), using 10 μl of the bead suspension in a 50 μl reaction with NEBNext Q5 2X PCR Master Mix (NEB) and 0.5 μM each Hi-C forward and reverse primers (Eurofins; Hi-C fwd: AATGATACGGCGACCACCGA, Hi-C rev: CAAGCAGAAGACGGCATACGA). Libraries were collected on a magnet and purification of the amplified DNA was carried out with magnetic beads (Agencourt AMPure XP) in a ratio of 1:1.6. Purified samples were eluted in 20 μl of EB. Quality of the generated Hi-C-libraries was analyzed using the High Sensitivity D1000 ScreenTape Kit (Agilent) and concentration was determined using the Qubit High Sensitivity dsDNA Kit (Thermo Fisher Scientific). Libraries were sequenced 42 bp paired-end on a Illumina NextSeq550. Sequencing libraries are listed in Supplementary Table 6.

**Immunoblotting**. For immunoblotting of whole-cell extracts, $2 \times 10^6$ cells were lysed in 120 μL 2x SDS-Lysis Buffer (20% Glycerin, 4% SDS, 10% 2-mercaptoethanol, 0.107 M Tris pH 6.8, 0.02% bromophenol blue) and heated at 95 °C for 5 min. SDS-PAGE was performed using a 10% polyacrylamide gel (Biorad #4561035) at 120 V for 1 h and protein was semi-dry blotted on a PVDF membrane at 11 V for 1 h. Blots were blocked for 1 h at RT using 4% milk powder and antibody incubation was conducted overnight at 4 °C using anti-CTCF (CST, #2899) at 1:500, or anti-RAD21 (Abcam, ab992) at 1:1000 dilution. Following five washing steps, goat-anti-rabbit-HRP conjugated secondary antibody (Dako P0448) was added in a 1:2500 dilution and incubated for 1 h at RT. After washing, bands were detected using ECL reagent (Amersham) and Fusion Pulse Detection System (Vilber). Blots were stripped using 1:10 diluted ReBlot Plus mild solution (Merck). Actin was stained with anti-actin (Sigma-Aldrich, A2066) antibody in 1:2500 dilution for 30 min at RT and detection was performed as described above. Densitometric analyzes were conducted in ImageJ V.1.51.

**ChIP-seq library preparation**. Chromatin immunoprecipitation (ChIP) was performed in biological replicates of three independent donors as described previously with slight modifications[62]. Chromatin for all ChIP-seq experiments of siRNA-transfected cells was harvested 7 days after transfection, whereas MO controls were directly harvested after purification. Briefly, for H3K27ac, CTCF and RAD21 ChIP-seq, cells were crosslinked with 1% formaldehyde for 10 min at room temperature and the reaction was quenched with glycine at a final concentration of 0.125 M. Chromatin was sheared using sonication (Branson Sonifier 250) to an average size of 250 − 500 bp. A total of 2.5 μg of antibody against H3K27ac (Abcam, ab4729), CTCF (CST, #2899), or RAD21 (Abcam, ab992), was bound to 20 μl pre-washed Dynabeads Protein A (Thermo Fisher Scientific) in a total volume of 200 μl PBS containing 0.02% Tween-20 for 1 h at room temperature rotating at 6 rpm. After, sonicated chromatin was added to the antibody-coupled Dynabeads (sonicated chromatin of approx. $1.7 \times 10^6$ cells) and incubated for 3 h at room temperature rotating at 6 rpm. Beads were washed on a magnet and chromatin was eluted. After crosslink reversal, RNase A, and proteinase K treatment, DNA was extracted with the Monarch PCR & DNA Cleanup kit (NEB). Sequencing libraries were prepared with the NEBNext Ultra II DNA Library Prep Kit for Illumina (NEB) according to manufacturer's instructions. The quality of dsDNA libraries was analyzed using the High Sensitivity D1000 ScreenTape Kit (Agilent) and concentrations were assessed with the Qubit dsDNA HS Kit (Thermo Fisher Scientific). Libraries were single-end sequenced on a NextSeq550 (Illumina). Sequencing libraries are listed in Supplementary Tables 7 (H3K27ac), 8 (RAD21), and 9 (CTCF).

**ATAC-seq**. ATAC-seq was carried out as described before[62]. For robustness, three biological replicates comprising three independent donors were used for each condition. Briefly, cells were harvested after 7 days of culture and treated in culture medium with DNase I (Sigma) at a final concentration of 200 U/ml for 30 min at 37 °C prior to transposition. After DNase I treatment, cells were washed twice with ice-cold PBS and cell viability and the corresponding cell count was assessed. For each ATAC reaction $5 \times 10^4$ cells were aliquoted into a new tube and spun down at $500 \times g$ for 5 min at 4 °C, before the supernatant was discarded completely. The cell pellet was resuspended in 50 μl of ATAC-RSB buffer (10 mM Tris-HCl, pH 7.4, 10 mM NaCl, 3 mM MgCl2) containing 0.1% NP-40 (0.01% for MO samples), 0.1% Tween-20 and 1% digitonin (Promega) and incubated on ice for 3 min to lyse the

cells. Lysis was washed out with 1 ml of ATAC-RSB buffer containing 0.1% Tween-20. Nuclei were pelleted at $500 \times g$ for 10 min at 4 °C in a fixed angle centrifuge. The supernatant was discarded carefully, and the cell pellet was resuspended in 50 µl of transposition mixture (25 µl 2x TD buffer, 2.5 µl transposase (100 nM final; Illumina), 16.5 µl PBS, 0.5 µl 1% digitonin, 0.5 µl 10% Tween-20, 5 µl H2O) by pipetting up and down 6 times. The reaction was incubated at 37 °C for 30 min with mixing (1000 rpm), before the DNA was purified using the Monarch PCR & DNA Cleanup Kit (NEB) according to the manufacturer's instructions. Purified DNA was eluted in 20 µl EB, and 10 µl the purified sample was objected to a 10 cycle PCR amplification using Nextera i7- and i5-index primers (Illumina). Purification and size selection of the amplified DNA was carried out with magnetic beads (Agencourt AMPure XP). For purification the ratio of sample to beads was set to 1:1.8, whereas for size selection ratio was set to 1:0.55. Purified samples were eluted in 15 µl of EB. The quality and concentration of the generated ATAC-libraries were analyzed using the High Sensitivity D1000 ScreenTape Kit (Agilent) and libraries were sequenced paired-end on a NextSeq550 (Illumina). Sequencing libraries are listed in Supplementary Table 10.

**RNA-seq library preparation**. Total cellular RNA was isolated from MO, MAC, and moDC (untreated, mock-, LPS, or siRNA-treated) using the RNeasy Mini Kit (Qiagen) according to the manufacturer's instructions. The concentration and quality of the purified RNA was analyzed using the RNA ScreenTape Kit (Agilent). Generation of dsDNA libraries for Illumina sequencing from total cellular RNA was carried out using the TruSeq Stranded Total RNA Kit (Illumina) or the ScriptSeq Complete Kit (Illumina) according to the manufacturer's instructions. The quality of dsDNA libraries was analyzed using the High Sensitivity D1000 ScreenTape Kit (Agilent) and concentrations were assessed with the Qubit dsDNA HS Kit (Thermo Fisher Scientific). Sequencing was performed using the Illumina NextSeq550 sequencer and libraries are listed in Supplementary Tables 11 and 12. Accession numbers of published CAGE-seq data for human MO, moDC, and MAC[63] are listed in Supplementary Table 13 and additional published human RNA-seq data for human MO, moDC, and MAC[27] are listed in Supplementary Table 14. Published and reanalyzed RNA-seq data for mouse bone marrow-derived macrophages (wildtype and *Rad21*-deficient, LPS stimulated and unstimulated)[16] are listed in Supplementary Table 15.

**Hi-C analysis**. The analysis of Hi-C data was primarily performed using the pipeline implemented in the HOMER package (v.4.11) as described in[64]. Samples and total read coverage are summarized in Supplementary Table 6. Raw sequencing reads were trimmed using homertools with options "trim −3 GATC -matchStart 20 min 20" before individual alignment to the reference chromosomes of the human genome (GRCh38.p10, release 27, www.gencodegenes.org/) using bowtie2. Mapped reads were paired and collected in HOMER style TagDirectories for further analysis. Filtered TagDirectories were generated using options "-tbp 1 -restrictionSite GATC -both -genome hg38 -removePEbg -removeSelfLigation -removeSpikes 10000 5". QC measures including the fractions of unique read pairs after filtering of local interactions, fractions of interchromosomal read pairs, total contacts after filtering and fractions of paired-end reads across distances were determined by HOMER are plotted in Supplementary Figs. 1i-k, 5c, d. For downstream analyzes, filtered TagDirectories of control cell populations (mock, siCTRL) or siRNA treatments (RAD21 or CTCF) were combined for each donor. For the initial detection of inter-chromosomal interactions, we merged donor TagDirectories. The analyzeHiC program of HOMER was then used to detect significant interchromosomal interactions of each cell type at a resolution of 2.5 Mb. Log2 ratios of interchromosomal interaction matrices of MO and MAC (as shown in Fig. 1h) were calculated donor-wise in R, averaged across donors and plotted using the image function in R. Pearson correlations across merged significant interactions of individual cell types and donors were calculated in R and correlation matrices shown in Fig. 1i and Supplementary Fig. 5f were clustered and plotted using the heatmap.2 function in R. PCA analysis on interaction matrices was performed using HOMER's runHiCpca.pl program with options "-res 25000 -window 50000 -genome hg38" and filtered TagDirectories. PC1 tracks as shown in Figs. 1j, 5c and Supplementary Fig. 1l, 5e represent averages of three donors. Hi-C contact maps for chromosomal regions as shown in Figs. 1j, 3c, d, 6a and Supplementary Figs. 1l, 3d, 6a were generated donor-wise using the batch-MakeHiCMatrix.pl program and parameter "-split" or in Figs. 7d and 8c using parameters "-split -rotate -frac .5", averaged across donors and plotted using the image function in R. The multidimensional scaling (MDS) plot of PC1 data shown in Fig. 1k was generated using the plotMDS function in edgeR. HOMER's findTADsAndLoops.pl program and parameters "-find -res 2500 -window 10000" and "-find -res 3000 -window 15000" were used to initially identify TADs and Loops for merged data sets of individual cell types. TADs and loops were merged using HOMER's merge2Dbed.pl function and then scored for interactions in replicate donor samples using findTADsAndLoops.pl and the option "-score". For comparisons of siRNA-treated samples, we also calculated scores using the parameter "-normTotal 500000000" to normalize for total interactions. Size factors for normalization in DeSeq2 (v2.1.30.0) were then calculated by dividing total scores from scoring without interaction normalization by total scores derived from interaction normalized scoring. MDS plots of loop and TAD scores in Figs. 2c, 3e, 5e used normalized, log-transformed, batch-corrected scores (using the removeBatchEffect function in limma (v3.46.0)) and were generated using the plotMDS function in edgeR. To identify

differential loops and TADs between control samples (MO, moDC, and MAC), we performed pairwise comparisons of replicate donor samples by first generating merged loop and TAD files for each pair, scoring them as described above and then determine differential TADs or loops using replicate data and HOMER's getDiffExpression.pl program. Corresponding analyzes of siRNA-treated samples were done using Deseq2 and normalization to total interactions using size factors as described above. Loop set enrichment analyzes shown in the bottom panels of Figs. 7f, 8f, and Supplementary Figs. 7e, 8d were performed using the fgsea package (v1.16.0) in R, and corresponding bar code representations were plotted using the barcodeplot function of the limma package (v3.46.0). Overlaps between genomic regions (including TADs, loop domains, loop anchors, ChIP-seq or ATAC peaks, genes, etc) were determined using the Bedtools suite (v2.27.1). Distributions of loop sizes (as shown in Fig. 5g and Supplementary Fig. 5i) were visualized using the beanplot package in R. Venn diagrams as shown in Figs. 4a, 6h, and Supplementary Figs. 4a, b, 6h were drawn in R using the venneuler package and formatted in Adobe Illustrator (v25.2.1). Histograms of insulation scores (Ins) and directionality indices (DI) as shown in Figs. 2b, d, 3f, 5f, 6g and Supplementary Figs. 2c, d, 3e, f, 5h & 6g were calculated using HOMER's annotatePeaks.pl function and parameters "-size 200000 -hist 250" using Ins- and DI-bedgraph files generated during TAD and loop finding using the findTADsAndLoops.pl program and plotted in R.

**ChIP-seq analysis**. Reads (single-end) were aligned to the human genome (GRCh38.p10) using bowtie2 in very sensitive mode. Lower quality alignments were removed by filtering reads with mapQ = <10 Only reads mapping to single unique locations and mapQ = >10 were included into HOMER style tagDirectories (using the option -unique in the makeTagDirectory program). Initial quality control was performed by calculating the fraction of reads in peaks (FRIP, summarized in Supplementary Tables 7–9) by running HOMER's (v4.11) findPeaks program in "factor" or "histone" mode using default parameters and the appropriate matching background data set (either ChIP input, genomic DNA or control ChIP). For further analyzes, chromosome scaffolds were removed. TF ChIP-seq peaks were called using HOMER's findPeaks program in "factor" mode with -fdr 0.00001 to identify focal peaks. Peak sets were filtered by subtracting blacklisted genomic regions, and by filtering out regions with a mappability <0.8. The latter was annotated to peak regions from mappability tracks generated with the GEM package using HOMER's annotatePeaks.pl. Multidimensional scaling (MDS) plots of H3K27ac ChIP-seq data sets (Figs. 1b, 7a and Supplementary Fig. 7a) used batch-corrected, normalized log-transformed count data (using the removeBatchEffect function in limma (v3.46.0)) and were generated using the plotMDS function in edgeR. Statistically significant differences in read counts across regions between sets of replicate H3K27ac ChIP-seq experiments were determined by merging peak sets of the compared samples (using the mergePeak function in HOMER) and applying edgeR (v3.32.1) with quantile (0.95) regression, GC and length correction performed using the cqn package in R (4.0.3). To identify super enhancers (SE), H3K27ac peaks were merge if they were less than 12 kb apart and if their centers were at least 2 K away from annotated TSS (gene annotation from GENCODE 44, release 27). Merged enhancers were ranked based on input-corrected read counts (generated using HOMER's annotatePeaks.pl function). The cut-off for SE was determined by using a tangential diagonal and SE count and cut-off are reported along with typical hockey-plots (as shown in Supplementary Fig. 1g). Statistically significant differences in read counts across SE were calculated in edgeR using the using the quasi-likelihood F-test. The multidimensional scaling (MDS) plot of SE data sets (Fig. 1b) used batch-corrected, normalized log-transformed count data (using the removeBatchEffect function in limma (v3.46.0)) and were generated using the plotMDS function in edgeR. Statistically significant differences in read counts across peaks between sets of replicate CTCF or RAD21 ChIP-seq experiments were determined using HOMER's getDiffExpression.pl with parameters "-peaks -batch -rlog -norm2total", which implements DeSeq2 (v2.1.30.0). Peak set enrichment analyzes were performed using the function fry of the limma package (v3.46.0) in R, except for comparisons of CTCF or RAD21 ChIP-seq experiments (Fig. 6c, e, and Supplementary Fig. 6c, e), which were analyzed using the fgsea package (v1.16.0) in R. Corresponding bar code representations (as shown in Figs. 4c, e, g, 6c, e, and Supplementary Figs. 2e, f, 6c, e) were plotted using the barcodeplot function of the limma package (v3.46.0), and bar plots of adjusted enrichment P values were plotted using standard functions in R (Figs. 3h, i, 4d, f, h and Supplementary Fig. 2e, f). Read coverage across individual peaks sets (as shown in Figs. 2a, 3a, b, 4a, 5b, 6b, d, 8k and Supplementary Figs. 4a, b, 5b, 6b, d, and 8i) was calculated using HOMER's annotatePeaks.pl with parameters "-hist 25 -ghist" using averaged replicate data sets and plotted in R using the image function. For small histograms presented on top of these plots in Figs. 4a, 5b, 6b, d, and 8k and Supplementary Figs. 4a, b, d-f, 5b, 6b, d, 7b, and 8i, the average read coverage was calculated with HOMER's annotatePeaks.pl with parameter "-hist 25" and plotted using basic R plotting functions. For histograms in Fig. 1d, average read coverage data and 95% confidence intervals were calculated in R and the ggplot2 package was used to draw histograms. Peak positions and results from differential peak calling are provided in Supplementary Data files 1–4.

**ATAC-seq analysis**. Reads (paired-end) were aligned to the human genome (GRCh38/hg38) using bowtie2 in very-sensitive and no-discordant modes, keeping only reads that map to a single unique genomic location for further analysis

(MAPQ > 10). Read positions were adjusted to move the ends proximal to the Tn5 binding site (for reads on the positive strand, the start is shifted +4 bp and its partner reads start −5 bp, for reads on the negative strand, the start is shifted −5 bp and its partner reads start +4 bp). Initial quality control was performed by calculating the fraction of reads in peaks (FRIP, summarized in Supplementary Table 10) by running HOMER's findPeaks program using parameters "-region -size 150". ATAC-seq peak regions were called from merged replicate data by combining two different approaches: The basic peak region set was called using HOMER's findPeaks program in 'region' mode using parameters "-size 150 -minDist 250 -L 2 -fdr 0.00001" to identify regions of variable length by stitching nucleosome-size peaks. To exclude shallow peak regions, only those were kept that overlapped a second peak set that was generated in "factor" mode using parameters "-size 250 -minDist 250 -L 2 -fdr 0.00001" to identify focal peaks. Statistically significant differences in read counts across peaks between sets of replicate ATAC-seq experiments were determined with quantile (0.95) regression and GC correction using edgeR (v3.32.1) with the cqn package in R (4.0.3). Read coverage across individual peaks sets (as shown in Figs. 2a, 3a, b, 4a, 5b, 6b, d, 7b, 8a, k and Supplementary Figs. 4a, b, d-f, 5b, 6b, d, 7b and 8a, i) was calculated using HOMER's annotatePeaks.pl with parameters "-hist 25 -ghist" using averaged, normalized replicate data sets and plotted in R using the image function. For small histograms presented on top of these plots in Figs. 4a, 5b, 6b, d, 7b, 8a,k and Supplementary Figs. 4a, b, d-f, 5b, 6b, d, 7b, and 8a, i, the average read coverage was calculated with HOMER's annotatePeaks.pl with parameter "-hist 25" and plotted using basic R plotting functions. For histograms in Fig. 1d, average read coverage data and 95% confidence intervals were calculated in R and the ggplot2 package was used to draw histograms.

**Motif analysis**. De novo motif discovery in peaks or regions (as shown in Figs. 4a, 7b, 8a, k, and Supplementary Figs. 1e, f, 7b, 8i) was performed with HOMER's findMotifsGenome.pl program and parameters "-len 7,8,9,10,11,12,13,14 -h". For searches in ChIP-seq peaks we used a 200 bp, peak-centered window, while for differential ATAC regions the given region sizes were used. For comparative motif analyzes, signature motifs were annotated using the findMotifsGenome.pl program and '-known'. Balloon plots of motif enrichment significance levels shown in Figs. 1f, 4b, 8b, l and Supplementary Figs. 2b, 3a, 8j were generated using the ggplot2 (v3.3.3) package in R. To determine peak-wise motif co-association we first performed a known motif search using HOMER's findMotifsGenome.pl across peak sets and determined the list of known motifs overlapping the previously determined de novo motif classes (e.g. Ebox, ETS, GATA, or RUNX). All listed motifs were then counted in peak regions using HOMER's annotatePeaks.pl with parameters "-mknown.motifs -matrixMinDist 4 -nogene -noann -nmotifs". Motif overlap in each individual peak was then reduced to motif class overlap (using the filtered known motif list), which was counted as positive for a particular class, if one of the class matching known motifs was present, or negative, if none was present. The count table was then used to generate a motif co-occurrence matrix and to calculate node sizes and edges width (each represented as % of all peaks). Networks of motif co-association (as shown in Fig. 1g) were generated in R using the igraph package (v1.2.5). To improve the visualization, colors of individual nodes were edited in Adobe Illustrator (v25.2.1). To determine the number and orientation of CTCF motifs across loop anchors we used a 20 bp CTCF motif (de novo derived from MAC ChIP-seq data) with the annotatePeaks.pl program. The stacked bar plot shown in Supplementary Fig. 3a was drawn in R using standard plot functions. Motif scores for CTCF were annotated to CTCF peaks using HOMER's annotatePeaks.pl with parameter "-mscore". Distributions of motif scores (as shown in Fig. 6f and Supplementary Fig. 6f) were visualized using the beanplot package in R.

**RNA-seq analysis**. Sequencing reads were mapped using STAR (v2.5.3a). For human samples we mapped to GRCh38.p10 and the genome index incorporated gene annotation from GENCODE 44 (release 27) to aid in spliced alignment. For the re-analysis of published mouse data, we mapped to GRCm38.p5 (GENCODE release M16) with the corresponding gene annotation. Tables of raw uniquely mapped read counts per human gene were generated during mapping using the built-in -quantMode GeneCounts option in STAR. Samples and basic QC data are summarized in Supplementary Tables 11, 12, 14, and 15. Differential expression analysis was carried out on raw gene counts using edgeR 3.32.1 in R (v4.0.3). Pairwise comparisons of indicated data sets were done using the quasi-likelihood F-test (glmTreat function in edgeR) against a given fold-change threshold (threshold for cell type comparisons: 2-fold change; threshold for knockdown comparisons: 1.5-fold change, threshold for volcano plots: zero). Multidimensional scaling (MDS) plots (Figs. 1b, 7a and Supplementary Fig. 7a) used batch-corrected, normalized log-transformed count data (using the removeBatchEffect function in limma (v3.46.0)) and were generated using the plotMDS function in edgeR. Volcano plots of edgeR results in Figs. 7c, 8d and Supplementary Figs. 3g, 7c, 8b were generated using the ggplot2 (v3.3.3) and ggrepel (v0.9.1) packages in R and labels were edited in Adobe Illustrator (v25.2.1). Heatmaps of differentially expressed genes shown in Fig. 9c and Supplementary Figs. 1a, 9d, e, used log2-transformed, batch-corrected, normalized and scaled CPM data (counts per million, Z scores were calculated using the scale function in R) and were generated using the

heatmap.2 function of the gplots package in R. Gene set enrichment analyzes using individual gene sets were performed using the function fry of the limma package (v3.46.0) in R. Corresponding bar code representations (as shown in Figs. 2e, 3g, 4i, 7e, f, g and 8e–h and Supplementary Figs. 7d–f, 8c–f, 9a, b) were plotted using the barcodeplot function of the limma package (v3.46.0), and bar plots of adjusted enrichment P values were plotted using standard functions in R (Figs. 2e, 3g, 4j). Gene set enrichment analyzes using gene sets in the ImmuneSigDB subset of the MSigDB[65] (retrieved using the msigdbr package (v 7.2.1) in R) were performed using the function camera of the limma package (v3.46.0) in R. Corresponding bar code representations (as shown in Supplementary Fig. 1b, c) were plotted using the barcodeplot function of the limma package (v3.46.0). For the presentation of expression levels of selected genes as shown in Fig. 1c, Fig. 9d, and Supplementary Figs. 2a, 9c, f, normalized and batch-corrected expression data were corrected for transcript length and plotted using the ggplot2 (v3.3.3) package in R. Statistically significant enriched KEGG pathways were identified using the kegga function in limma (v3.46.0) and balloon plots of significance levels shown in Figs. 7h, 8i and Supplementary Figs. 7g, 8g were generated using the ggplot2 (v3.3.3) package in R. The scatter plot shown in Fig. 9b was also generated using the ggplot2 (v3.3.3) package in R. For comparisons of human and mouse expression data (as presented in Supplementary Fig. 9d, e), orthologous gene pairs were identified using the getLDS function of the biomaRt (v2.46.1) package, and data was scaled independently for each species.

**Generation of read coverage tracks**. HOMER was used to generate sequencing-depth normalized bedGraph/bigWig files of ChIP-seq and ATAC-seq data (using standard parameters for ChIP and a fixed fragment length of 65 bp for ATAC). For visualization purposes and equivalent to the normalization of read count data, ATAC-seq and H3K27ac ChIP seq tracks from individual donors were scaled based on the top ~5% peaks (5000 or 3000 peaks, respectively, no batch correction applied) across all samples and normalized BigWigs files from replicate data sets were averaged using the program bigWigMerge and dividing the count data by the number of samples. Sequencing-depth normalized bedGraph/bigWig files of RAD21 and CTCF ChIP-seq runs were averaged as above (without prior scaling), except for the knockdown data sets, which were merged on tagDirectory level before the generation of sequencing-depth normalized bedGraph/bigWig files. BedGraph files were converted to BigWig using the program bedGraphToBigWig. Tracks and loop data for selected regions as shown in Figs. 1e, j, 3c, d, 6a, 7d, 9c and Supplementary Figs. 1d, h, 3d, h, 4h–k, and 6a, were visualized using the pyGenomeTracks software (v3.5). Figures were assembled and formatted in Adobe Illustrator (v25.2.1).

## Data availability

The raw sequencing data generated in this study are deposited with EGA (study accession number is EGAS00001005508. The human sequencing data are available under restricted access and can be obtained by qualified researchers. Processed data files (bigwig tracks and peak files for ATAC and ChIP, interactions for HiC data and read count tables for RNA-seq data) are deposited with ArrayExpress (accession numbers: E-MTAB-10844, E-MTAB-10845, E-MTAB-10846, E-MTAB-10848, E-MTAB-10849). [https://www.ebi.ac.uk/arrayexpress/experiments/E-MTAB-XXXX/, etc.]. The source data underlying Figs. 1c, f, g, i, k, 2b, d, e, 3f–j, 4a, b, d, f, h, j, 5f, g, 6b, d, f–h, 7b, e, f, h, 8a, b, i, k, l, 9b–d and Supplementary Figs. 1a–c, e, f, i, k, 2a–f, 3a–c, e–g, 4a–g, 5a, c, f, h, i, 6b, d, f–h, 7b, d, g, 8g, i, j and 9c–f are provided as a Source Data file. Source data are provided with this paper.

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

## Acknowledgements

We thank Sven Heinz (UCSD, San Diego) for sharing protocols. Hi-C-, ChIP-, ATAC- and RNA-sequencing were conducted at the NGS Core of the Leibniz Institute for Immunotherapy (LIT, University Regensburg and University Medical Center Regensburg, Germany). This study was funded by a grant of the Else Kröner-Fresenius-Stiftung to M.R. (2014_A223), and a SPP2202 Priority Program grant to C.G. by the Deutsche Forschungsgemeinschaft (GE 3202/1-1).

## Author contributions

M.R. designed the study with contributions by J.M., C.G., and A.F.; J.M. performed most experiments with contributions from A.F., K.Ma., K.Me., M.N., J.R., H.S., U.A., and C.G.; R.M. contributed to data acquisition; M.R. analyzed sequencing data with help from J.M.; M.R. wrote the original draft with contributions from all authors.

## Funding

## Competing interests

The authors declare no competing interests.
