## [Peer Review File · Nature Communications]

Postmitotic differentiation of human monocytes requires cohesin-structured chromatinREVIEWER COMMENTS

Reviewer #1 (Remarks to the Author):

This well-written and interesting article uses the post-proliferation (cell-cycle independent) differentiation of primary human monocytes in vitro as a model to demonstrate, by siRNA depletion, the contribution of cohesin to the active transcription programs of differentiation and activation of macrophages.

The authors conclude that their technical findings confirm the importance of cohesin to the architecture of global transcriptional control of monocyte programs and to haematological disease. Cohesin is a structural protein that binds DNA and controls DNA structure for replication.

At a local gene level, cohesin controls the structure of DNA loops for transcription; studies show local cohesin forming DNA loops to control the expression of individual genes.

The authors aim to show that global chromatin architecture controlled by cohesin is important for broader differentiation programs.

For this, they use a model system of human monocyte that can differentiate without DNA replication (ie avoiding the confounder of cohesin linked to DNA replication).

Minor suggestions:

1. The chosen methods for monocyte differentiation into DCs and Macs are good models to monitor expression programs. However, GM-CSF still can induce monocyte proliferation.
2. The authors should add a note of caution that the monocytes used as controls were already mock treated or electroplated with control siRNA, and a comment that this was a control for the procedures and if this might have affected the results.

Reviewer #2 (Remarks to the Author):

The manuscript by Minderjahn describes 3D conformational features of human primary monocytes (MO), macrophages (MAC) and dendritic cells differentiated from MO in vitro (MAC were examined more thoroughly than DCs in this work). The authors performed methodical analyses of Hi-C, along with ATAC-seq and H3K27ac ChIP-seq. Extensive changes were observed in chromatin architecture (Hi-C) along with chromatin accessibility and H3K27ac enhancer landscape after differentiation.

These changes coincided with changes in transcriptome profiles revealed by RNA-seq. These analyses were systematic, and included PCA and global, analysis of inter-chromosomal architectures, intrachromosomal TADs and loops. To delineate the mechanistic basis of chromatin architecture formation, the authors knocked down (KD)RAD21 (part of Cohesin complex) and CTCF. The KD of either factor led to a reduction in TADs and loops and resulted in reduction in ATAC peaks and H3K27ac peaks, indicating that 3D conformation is important for setting epigenome structure relevant to gene expression. Supporting this idea, RNA-seq showed that MAC transcriptome profiles are markedly altered upon KD of RAD21 or CTCF. RAD 21 KD also led to extensive dysregulation of LPS induced gene expression in MAC. From these results, the authors conclude that 3D chromatin architecture plays an essential role in transcriptional programs of MO during differentiation and LPS mediated inflammatory response.

General comments: This is a large, systematic analysis of 3D conformation during macrophage differentiation and inflammatory activation. The most significant aspect of this work is that (a) MO-MAC differentiation coincides with extensive changes in 3D conformation (TADs, loops), which correlate with changes in gene expression, (b) RAD21 KD significantly affects gene expression profiles during MAC differentiation, and (c) RAD21 KD alters LPS induced inflammatory transcriptome profiles. Techniques and data analyses used in this work are at the cutting edge and sound. Overall, the data support the authors' view that chromatin architecture undergoes dynamic changes during macrophage differentiation (and LPS stimulation), which corresponds to changes in gene expression. This study provides a new insight into the relation between Hi-C landscape and innate immune activities. Thus, this reviewer is, in general, in favor of potential publication in Nat. Commun. One caveat is that while there is a good correlation between conformational changes and changes in gene expression, direct causality is not proven (some TADs/loops changes do not

correlate with gene expression/differentiation, and vice versa). Also, figures correlating Hi-C data and transcriptome data lack clarity. It would be important to describe more precisely the role of conformation in gene expression.

Specific comments: 1. Data in Fig. 2d, 2e lower panels apparently present CAGE-seq data, correlating Hi-C and RNA-seq data. However, no explanations are provided regarding CAGE analysis. CAGE-seq is not mentioned in the Methods section either. It is important to disclose authors methodology more completely. The summary figures in 2g and 2h also lack clarity. It will be more convincing if the authors provide full description of the lower panels. The diagrams in the upper panels are equally unclear (the same for Fig. 3h)

2, Fig. 3h is a summary diagram explaining all Fig 3 data. However, the legends for this panel is too cursory: changes in loops in the left and H3K27ac frags in the right are not explained, for example. Was CAGE-seq used to derive this? The summary sentence in p. 7 does not directly refer to Fig 3h. It would help the readers to understand the figure more fully, if the authors explain the figure, telling the meaning of gain of structural loops, and the meaning of big and small arrow and H3K27Ac frags.

3. Hi-C was performed for MAC with Rad21 KD and CTCF KD. Fig 4d shows that ATAC-seq and H3K27ac peaks are reduced, but to a lesser degree relative to the reduction of TADs and loops. This would suggest that some of 3D architectural changes do not directly correlate with chromatin accessibility and or enhancer configuration (relevant to transcription). Again, it will be helpful if the authors provide the meaning of these observations.

4. The example of the LPL locus in Fig. 4h is beautiful, but this is only one aspect of the complex changes occurring in Hi-C landscapes. It might be more revealing to characterize TADs and loops linked to the reduction on ATAC-seq and H3K27ac. Are TADs/loops associated with reduced ATAC peaks overlap with those for H3 K27ac? Are the peaks specific for MAC or common with MO? What are ATAC peaks and H3K27ac peaks unaffected by RAD21KD? Are they located within or outside of . TADs/loops?

5. Fig. 7. The data in this figure are important and interesting, since they show that RAD21KD significantly alters LPS induced transcriptome profiles in MAC. The results suggest that the LPS signal changes 3D landscape within 4 h to initiate timely innate immune responses. However, it is somewhat disconcerting that no Hi-C data are presented for LPS stimulated MAC. Fig. 7 depicts changes in RNA-se profiles in control and RAD21 KD after LPS treatment. It is imperative to present (a) increase/decrease in TADs/loops after LPS stimulation, (b) the relation to ATAC/H3K27ac peaks (as in Fig 3, and Fig. 4) and (c) the effects of RAD21 KD). A more extensive correlation study may be warranted to relate LPS induction (repression) of gene expression with LPS induced changes in Hi-C. I would emphasize that elucidating the functional role of Hi-C conformation is highly consequential, because LPS induces inflammatory responses not only in human macrophages studied here, but widely in myeloid cells of many other species and LPS-induced inflammation is relevant to various diseases.

6. Assuming that LPS stimulation induces changes in chromatin conformation, the authors should comment on possible mechanisms by which LPS treatment can change chromatin architecture so rapidly.

7. LPS induction of inflammatory genes highly transient, in that transcription ceases within 24 h for the most part. Are LPS induced TADs/loops .similarly transient?

Reviewer #3 (Remarks to the Author):

This study presents a comprehensive analysis of the chromatin changes during human monocyte differentiation into macrophages and dendritic cells as well as assess the effects of Cohesin and CTCF depletion on this developmental progression.

The key findings in this study in human monocytes reflect observations previously seen of cohesin's role in regulation of cell-type specific gene expression, its importance in the differentiation in mouse monocytes and human HSPCs as well as in the regulation of immune response genes.

I have some major comments below I believe are important to provide a complete study.

Major points

- To assess the identity of derived cell-lines and with the availability of RNA-seq for these samples, a transcriptome wide comparison can be made. Markers of these cell lines can be derived from public databases and used as gene sets for a GSEA style of analysis against for the MO, MAC, moDC transcriptomes. This is especially important as the cell identity is central to many of the findings.

-- In addition a supplementary heat map of the these markers would provide a useful visual overview of the above and provide some understanding of the within sample variability.

-Global changes in genome wide accessibility are expected both over differentiation and , perhaps to a lesser extent, following cohesin depletion. The FRIP scores for ATAC-seq presented in supplementary tables as well as the ATAC-seq signal plots in figure 1 show a confounding of overall ATAC-seq signal with MO, MAC and moDC groups. The use of quantile normalisation in the identification of statistical differences across groups would assume much of this change in efficiency is technical whereas in this context it is may be a reflection of expected biology. A similar change in total RNA expression levels may be expected in conjunction with this change in overall accessibility. Studies like this have used external RNA spikes in for RNA-seq and normalisations to total mapped reads for ATACseq.

--The changes on global ATAC efficiency and their impact of differentially accessible sites should be evaluated.

--For the RNA-seq, comments on use of no-spikes in comparison to previous studies can be made.

-The identification of MO-specific loops which are CTCF independent suggests that another factor is perhaps acting at these sites. This is very interesting observation alongside previous work on de-novo loop formation in other systems. Previous work has identified monocyte to macrophage, esc to npc and epidermal differentiation loop formation may be less dependent on CTCF binding and associated with specific transcription factors.

--In this context, an analysis of the motifs and putative transcription factors at these sites would be important to this study.

--A comment on the loss of cohesin independent interactions in respect to whether these observations match any gain of cohesin independent denovo interactions along development in other cell types.

- Some more information on what the batches used in correction for visualisation were should be included in the methods. I am assuming patient in this case but this should be stated clearly as additional batch correction on unknown factors is often a problem for reproducibility.

- The effects on regulation of inflammation induced genes in cohesin depleted cells is interesting although ,as the author notes, opposed to that observed in mouse monocytes.

In order to better align the comparisons, a side by side heat map of the induced genes and gene sets from the Cuartero study (translated to human orthogues for the mouse data) would be very informative.

-Given the availability of H3K27ac in this study and since both the connectivity between super enhancers have been shown to be promoted by cohesin, and super enhancers are believed to play an important role in cell identity, I would be interested to hear comment on any effects on SEs over development and following cohesin depletion.

-Due to the complexity of bioinformatics analyses performed in this study, it is important to provide some processed results.

--Tables of differential analysis of ATAC, RNA and ChIP should be included.

--Tables of files of loop boundaries and the overlap with CTCF/Cohesin should be included

A few minor comments.

- It would be interesting to see the enriched motifs represented as a motif by sample heat map (such as implemented in ChromVar). This would allow for a visual assessment of the relative strength of enrichment for motifs across groups and the variability of enrichment within samples.

- Figure 2c could be presented as epigenetic heat map (as in 3f) of all the loop domains groups as illustrated in the the Figure 2c venn diagram to provide more resolution in the correlation of absolute ChIP signals to reported overlaps.

- Figures 2g and f, it is unclear what the multiple correction applied is and whether it is effected by how many tests were conducted at the same time (as in BH or bonferroni).

- "Genome ontology analysis" - I am unclear what this is and couldn't find the description in methods.

- Figure 2i is missing what the enrichment value is on y axis.

- Supplementary Figure 2i should point is now supplementary figure 2f I believe.
- Venn diagrams of overlaps (2c/3b) should be presented as tables in supplementary alongside odds ratios/p-values.
- "A large fraction of MO-specific loops lacked CTCF binding and instead were marked by H3K27ac, which was rarely seen in MAC or moDC-specific loops" should only point to Figures 3f and Supplementary Figures 3g-h.

- The observation of differential lengths for chromatin loops following cohesin depletion in MAC and moDC is interesting and in keeping with previous finding. Is there any potential mechanism why CTCF depletion shows a similar effect in MAC but not moDC cells?
- A supplementary epigenetics heat map to complement figure 4h showing CTCF and cohesin signal over strengthened and lost anchors would allow a reader to better assess the strength of peaks that overlap at these sites.
- I think Figure 5C would better fit a Chromvar heat map to show distribution of motif enrichment between conditions/cell types and variability of motif enrichment within samples.
- "Except for motif signatures (likely due to a higher background in moDC ATAC-seq data)" — How is higher back ground determined? From 1-FRIP?
- It is not clear to me how figure 5f and 6c were derived "KEGG pathway analysis of genes affected by CTCF loss in MAC".
- Why two different tests are needed for ChIP differential analysis, (for regions and peaks) is unclear to me.
- How are peaks defined for ATACseq/ChIPseq differential analysis over peaks? Were they reduced/collapsed into representative peaks such as implemented in DiffBind.
- The exact version of the GRCh38 genome used and a link to its source should be included in the supplementary for ease of reproducibility.
- The description in the text of the removal of multmapping reads after Bowtie2 alignment should be altered. Filtering by mapQ > 10 can include multi mapping reads and instead a strategy employing the filtering by XS SAM tag could be used. The text could be changed to "removal of lower quality alignments by filtering reads with mapQ =< 10". In my experience, this will not affect downstream analysis as mapQ reflects a scale of uniqueness in Bowtie2 alignment and removal of > 10 will indeed remove major artefacts but clarifying the methods for others wishing to follow is useful here.
- removeBatchEffect is a function in the Limma package, not edgeR.
- A supplementary heatmap showing the correlation between replicate ATAC-seq signals before and after batch correction would be helpful for the reader to assess inter patient variability and variability across replicates post batch-correction.
- "Statistically significant differences in read counts across peaks between sets of replicate ATAC-seq experiments were determined with quantile (0.95) normalization and GC correction using edgeR (v3.32.1) with the cqn package in R (4.0.3)". I am unsure if the replicates were used for this analysis or averaged for this test.

Response to reviewer's comments:

We would like to thank the reviewers for the time and effort they have invested in reviewing our manuscript. In response to their helpful comments and recommendations, we have revised the manuscript and hope to have fully addressed all the concerns and suggestions as detailed in our point-by-point description of the revision below.

In brief, the key changes to the manuscript are as follows:

1. We have added further analyses and interpretation regarding the specificity and causality of chromatin changes on gene expression. This also includes a complete overhaul of differential loop analyses in the knock-down settings.
2. We also made an effort to clarify cellular identities, as requested, and provide additional analyses to characterize each cell type.
3. We also looked into potential candidate factors that are either independent of CTCF/cohesin, or enriched in enhancers.
4. In addition we now also provide a rich set of Supplementary data, including positions of peaks, regions and domains, as well as results from analyses of differential gene expression or signals.

All changes in the main text are indicated by red lettering. Changes in figures include:

Fig.1b,d,f,g: Revised analyses of differential H3K27ac analyses, which were accidentally performed without batch-correction. Also added analyses and data regarding super-enhancers (based on H3K27ac data).

Supplementary Fig.1: Added additional analyses characterizing our differentiation model (new FigS1a-c). Revised the presentation of de novo motifs across differential H3K27ac regions in FigS1e (original FigS1b) and added corresponding data for differential ATAC peaks in new FigS1f. Also added hockey stick plots presenting super enhancer (SE) identifications (new FigS1g) and genome tracks for examples of SE regions (new FigS1h).

New Fig.2 and Supplementary Fig.2: Added revised and new analyses on TADs and their boundaries. Initially the threshold for differential TAD detection was $FC > 2$, which only resulted in very few differential TADs (original Fig.2b). After lowering this to $FC > 1.5$, we identified significant numbers of differential TADs and now show that TAD strength changes correlate with concordant changes in gene expression & (super)enhancer activity.

Revised Fig.3 (originally Fig.2): Replaced Venn diagrams of original Fig.2c with images showing the genomic distance distributions of RAD21, CTCF, H3K27ac, and ATAC coverage across feature-overlapping loop anchors (new Fig. 3a). A comparison of insulation scores and directionality indices was added in Fig. 3b. As a comparison, we also added corresponding images for regions overlapping CTCF or RAD21 that are not overlapping loop anchors. Original Fig.2b was revised and moved to new Fig.2. Original Fig.3a was revised and moved to revised Fig. 3d (numbers of anchors slightly changed, because we adjusted the filtering of overlapping anchors). We also slightly adjusted the assignment of features to loop domains (as outlined in the revised methods) and revised original Fig. 2f-h (now revised Fig. 3e,f) and added corresponding analyses for super enhancers in new Fig. 2g. We also replaced and labelled the schematic illustrations across all revised Figures, which hopefully makes them more comprehensible.

Revised Supplementary Fig.3 (originally Supplementary Fig.2): We added new analyses comparing insulation scores across various types of enhancers in new Fig. S3b. We also added new motif analyses of loop anchors that were neither associated with CTCF nor with RAD21

(new Fig. S3c). Original Fig. S2c,d were removed (summary statistics are shown in revised Fig.3) and histograms from original FigS3a,b were revised and moved to revised Fig. S3e,f.

Revised Fig.4 (originally Fig.3): Combined Venn diagrams of original Fig.3b and plots shown in original Fig.3e,f,h into a single revised figure that now also includes ATAC data (new Fig4a). We also added motif analyses of CTCF-independent loop anchors in new Fig.4a&b. Revised feature overlaps for RAD21 (original Fig. 3c&d, now Fig. 4c&d), CTCF (original Fig. 3i&j, now Fig.4e&f), added new analyses for H3K27ac or TSS activity at loop anchors in new Fig 4g-j. Original motif analyses in original Fig. 3g was replaced with more systematic analyses and presentation of motif signatures across differential RAD21 peaks in Figs.S4d-f. The summary scheme in original Fig.3h was replaced with a more detailed version in Fig.4k.

Revised Supplementary Fig.4 (originally Supplementary Fig.3): As in the revised main Fig. 4, Venn diagrams (original Fig.S3c), histograms (original Fig.S3f,i) and genomic distance distribution plots (original Fig.S3g,h) were combined into new Figs.S4a,b. Motif analyses of CTCF-independent loop anchors were added in new Fig.4a&c. Genomic distance distribution plots as well as motif analyses of differential RAD21 peaks were added in Fig.S4d-f. Original Figs.S3d,e,j,k were removed (summary statistics are shown in revised Fig.4) and example genome browser tracks for different types of loop changes are now shown in Fig.S4h-k.

Revised Fig.5 (originally Fig.4): Revised Fig.5a,b (original Fig.4a,b). In revised Fig.5e (original Fig. 4e), we added a panel showing insulation and directionality indices for TADs. Because we changed our strategy for differential loop analysis in knock-down settings essentially all Fig related to loops were revised. We added two MDS plots in new Fig. 5f showing results from differential TAD/loop analyses, revised Fig.5g (original Fig.4f), Fig5n (originally Fig.4i), Fig.5o (originally Fig.4h), and Figs.5j&l (original Fig.4j). Histograms in original Fig. 4k were replaced with more informative plots showing the genomic distance distributions of RAD21, CTCF, H3K27ac, and ATAC coverage across feature-overlapping, differential loop anchors (new Figs.5i,k). Also, we added a plot showing CTCF motif score distributions across differential anchors, which are different between strengthened and weakened loops.

Revised Supplementary Fig.5 (originally Supplementary Fig.4): Changes made in revised Fig.S5 for moDC data correspond to those made in revised Fig.5.

Revised Fig.6 (originally Fig.5): Differential ATAC coverage plot shown in original Fig. 5b was split into sites related to CTCF knock-down (revised Fig. 6b) and RAD21 knock-down (new Fig. 7a) which made it easier to discuss the results in the right context. We also added several new analyses addressing causalities, including examples for aberrant expression associated with boundary loss (new Fig. 6d), an analysis of genes that have promoter-bound CTCF (new Fig. 6d), as well as analyses addressing the behavior of housekeeping genes in CTCF knock-down (new Fig. 6f).

Revised Supplementary Fig.6 (originally Supplementary Fig.5): Changes made in revised Fig. S6 for moDC data correspond to those made in revised Fig.6.

Revised Fig.7 (originally Fig.6): Differential ATAC coverage plot shown in original Fig.5b was split into sites related to CTCF knock-down (revised Fig.6b) and RAD21 knock-down (new Fig.7a) which made it easier to discuss the results in the right context. We also added several new analyses addressing causalities, including examples for de novo loops created during MO differentiation that are lost upon RAD21 knock-down (new Fig.6c), analyses of genes within knock-down affected loops and TADs (new Figs.6e,f), as well as analyses addressing the behavior of housekeeping genes in RAD21 knock-down (new Fig.6d). Histograms in original Fig. 6d were replaced with more informative plots showing the genomic distance distributions of RAD21, CTCF, H3K27ac, and ATAC coverage across feature-overlapping, differential enhancers (new Fig.7k). Here, motif signatures were also added (new Fig.7k,l). Because it became difficult to summarize the findings in a single small image, we removed original Fig.6f.

Revised Supplementary Fig.7 (originally Supplementary Fig.6): Changes made in revised Fig.S7 for moDC data correspond to those made in revised Fig.7.

Fig.8 (originally Fig.7): Nothing changed.

Revised Supplementary Fig.8 (originally Supplementary Fig.7): Added new Figs.S8d-f, which compare gene expression changes of this study with a published data set on mouse macrophages.

Changes in the Supplementary data and tables include:

New Supplementary Tables 1-4: Counts and statistics for overlaps between accessible or H3K27ac marked sites and loop anchors.

New Supplementary Table 13: Accession numbers for published human CAGE-seq data used in this study.

New Supplementary Table 15: Accession numbers for published murine RNA-seq data of RAD21^{-/-} macrophages stimulated with LPS.

Supplementary Data File 1: Excel-File containing data related to MO, MAC and moDC: Results from differential gene expression analyses, ATAC peak positions, results from differential ATAC peak analyses, H3K27ac peak positions, results from differential H3K27ac peak analyses, super-enhancer (SE) positions, results from differential SE analyses.

Supplementary Data File 2: Excel-File containing data related to MO, MAC and moDC: Positions of TADs, results from differential TAD analyses, positions of loops, results from differential loop analyses, positions of RAD21 peaks, positions of CTCF peaks.

Supplementary Data File 3: Excel-File containing data related to siRNA-treated samples: Results from differential gene expression analyses, ATAC peak positions, results from differential ATAC peak analyses, H3K27ac peak positions, results from differential H3K27ac peak analyses, super-enhancer (SE) positions, results from differential SE analyses.

Supplementary Data File 4: Excel-File containing data related to siRNA-treated samples: Positions of TADs, results from differential TAD analyses, positions of loops, results from differential loop analyses, positions of RAD21 peaks, results from differential RAD21 peak analyses, positions of CTCF peaks, results from differential CTCF peak analyses.

Point-by-point response to referee comments:

Reviewer #1 (Remarks to the Author):

This well-written and interesting article uses the post-proliferation (cell-cycle independent) differentiation of primary human monocytes in vitro as a model to demonstrate, by siRNA depletion, the contribution of cohesin to the active transcription programs of differentiation and activation of macrophages. The authors conclude that their technical findings confirm the importance of cohesin to the architecture of global transcriptional control of monocyte programs and to haematological disease. Cohesin is a structural protein that binds DNA and controls DNA structure for replication. At a local gene level, cohesin controls the structure of DNA loops for transcription; studies show local cohesin forming DNA loops to control the expression of individual genes. The authors aim to show that global chromatin architecture controlled by cohesin is important for broader differentiation programs. For this, they use a model system of human monocyte that can differentiate without DNA replication (ie avoiding the confounder of cohesin linked to DNA replication).

We thank this reviewer for his/her appreciation of our work and the positive feedback.

Minor suggestions:

1. The chosen methods for monocyte differentiation into DCs and Macs are good models to monitor expression programs. However, GM-CSF still can induce monocyte proliferation.

We thank this reviewer for this comment. It may well be true that GM-CSF can induce proliferation in macrophage cultures (which we have not yet tested ourselves), but under moDC culture conditions (including GM-CSF, IL-4 and FCS) monocytes do not proliferate. We have shown this in a previous study (Klug et al 2010, Genome Biology) where we measured both the incorporation of [3H]-thymidine as well as 5-Bromo-2'-deoxy-uridine (BrdU) incorporation and found no evidence for DNA replication.

2. The authors should add a note of caution that the monocytes used as controls were already mock treated or electroporated with control siRNA, and a comment that this was a control for the procedures and if this might have affected the results.

We appreciate this comment but assume that this reviewer has overlooked the corresponding statements, which were already present in the original version of the manuscript. The procedure is stated in Figure 1a describing the experimental setup. It is clearly stated in the legend, that cells were electroporated because they also served as controls in siRNA experiments. We also mention this in the Methods (paragraph on Cells) together with the remark that we have not seen major changes in cell differentiation upon electroporation. But a corresponding reference has been missing, which we have now added.

Reviewer #2 (Remarks to the Author):

The manuscript by Minderjahn describes 3D conformational features of human primary monocytes (MO), macrophages (MAC) and dendritic cells differentiated from MO in vitro (MAC were examined more thoroughly than DCs in this work). The authors performed methodical analyses of Hi-C, along with ATAC-seq and H3K27ac ChIP-seq. Extensive changes were observed in chromatin architecture (Hi-C) along with chromatin accessibility and H3K27ac enhancer landscape after differentiation. These changes coincided with changes in transcriptome profiles revealed by RNA-seq. These analyses were systematic, and included PCA and global, analysis of inter-chromosomal architectures, intrachromosomal TADs and loops. To delineate the mechanistic basis of chromatin architecture formation, the authors knocked down (KD)RAD21 (part of Cohesin complex) and CTCF. The KD of either factor led to a reduction in TADs and loops and resulted in reduction in ATAC peaks and H3K27ac peaks, indicating that 3D conformation is important for setting epigenome structure relevant to gene expression. Supporting this idea, RNA-seq showed that MAC transcriptome profiles are markedly altered upon KD of RAD21 or CTCF. RAD 21 KD also led to extensive dysregulation of LPS induced gene expression in MAC. From these results, the authors conclude that 3D chromatin architecture plays an essential role in transcriptional programs of MO during differentiation and LPS mediated inflammatory response.

General comments: This is a large, systematic analysis of 3D conformation during macrophage differentiation and inflammatory activation. The most significant aspect of this work is that (a) MO- MAC differentiation coincides with extensive changes in 3D conformation (TADs, loops), which correlate with changes in gene expression, (b) RAD21 KD significantly affects gene expression profiles during MAC differentiation, and (c) RAD21 KD alters LPS induced inflammatory transcriptome profiles. Techniques and data analyses used in this work are at the cutting edge and sound. Overall, the data support the authors' view that chromatin architecture undergoes dynamic changes during macrophage differentiation (and LPS stimulation), which corresponds to changes in gene expression. This study provides a new insight into the relation between Hi-C landscape and innate immune activities. Thus, this reviewer is, in general, in favor of potential publication in Nat. Commun. One caveat is that while there is a good correlation between conformational changes and changes in gene expression, direct causality is not proven (some TADs/loos changes do not correlate with gene expression/differentiation, and vice versa). Also, figures correlating Hi-C data and transcriptome data lack clarity. It would be important to describe more precisely the role of conformation in gene expression.

We would like to thank this reviewer for his overall positive feedback. His main criticism, as we understand it, is our failure to provide evidence for direct causalities. The comment is well taken, and we were aware of this caveat (and had already discussed this in the original version of the manuscript (discussion, third paragraph)), but given the experimental design this is not straight forward. Our setup includes 7 days of culture, so it is rather difficult to distinguish between primary (causal) and secondary (indirect) effects of normal differentiation or upon cohesin/CTCF depletion. Additionally, siRNAs usually knockdown protein expression with a certain delay, which also need to be considered in the interpretation of our data. On top of this, HiC is not a very "precise" technology, which would allow the unambiguous assignment of loop anchors (which range between 10-20 kb) to individual regulatory or boundary elements (usually in the range of hundreds of bp) – its resolution is often too low, and regulatory loops are often too sparse to be reliably detected. Nevertheless, we have added more extensive explanations, and additional analyses, which aim at addressing causalities. E.g., we provide further evidence for concordant changes on the levels of TADs, loop domains and their anchors with gene expression and H3K27ac deposition at regulatory elements; we provide evidence that the functions of CTCF in gene regulation are likely related to its boundary function, but also to its independent function as a transcription or loop-directing factor; we more specifically characterize the effects of cohesin/CTCF depletion on different classes of genes (house keeping/differentiation associated); we also analysed CTCF motif affinities, which are clearly

lower at anchors that are affected by cohesin/CTCF depletion. As this reviewer suggested, we also included more explanations/interpretations to our findings already in the results section.

More definite analyses/statements about causalities would unfortunately require much more detailed studies looking into individual loci at a higher resolution (both temporal and spatial), which is clearly beyond the scope of this manuscript.

We regret that our previous data and schematic representations lacked clarity. As outlined below, we have also redesigned our explanatory charts accompanying conformation and transcription data to improve clarity. For studying the relationships between changes in chromatin interactions and transcriptional activity, we frequently make use of gene set enrichment analyses, which we think is the most appropriate type of test for our data sets. In the revised manuscript we now explain the rationale in more detail when we first use it (describing the new Figure 2), hoping that the interpretation will be more comprehensible.

Specific comments:

1. Data in Fig. 2d, 2e lower panels apparently present CAGE-seq data, correlating Hi-C and RNA-seq data. However, no explanations are provided regarding CAGE analysis. CAGE-seq is not mentioned in the Methods section either. It is important to disclose authors methodology more completely. The summary figures in 2g and 2h also lack clarity. It will be more convincing if the authors provide full description of the lower panels. The diagrams in the upper panels are equally unclear (the same for Fig. 3h)

We apologize for the lack of information regarding the CAGE data. The CAGE data for monocytes, monocyte-derived macrophages and dendritic cells were already published a few years ago (by us and our colleagues of the FANTOM consortium). We used it to display transcription output in genome tracks in combination with heatmaps presenting the conformation data, because CAGE data shows more focused signals at promoters compared to the RNAseq data. We have now added the corresponding information in Methods and added corresponding accession numbers in Supplementary Table 13. However, all statistical analyses were performed on the matched RNA-seq data available for each donor.

As also outlined below, we have redesigned and our explanatory charts accompanying conformation and transcription data to improve clarity.

2. Fig. 3h is a summary diagram explaining all Fig 3 data. However, the legends for this panel is too cursory: changes in loops in the left and H3K27ac frags in the right are not explained, for example. Was CAGE-seq used to derive this? The summary sentence in p. 7 does not directly refer to Fig 3h. It would help the readers to understand the figure more fully, if the authors explain the figure, telling the meaning of gain of structural loops, and the meaning of big and small arrow and H3K27Ac frags.

We thank the reviewer for pointing this out. We appreciate that this schematic required a better explanation. We have completely redesigned all explanatory charts and hope that the new design is now clearer (and better labelled). In addition, we have extensively revised the description and summary of our results. We hope that the changes made text and figures more understandable.

3. Hi-C was performed for MAC with Rad21 KD and CTCF KD. Fig 4d shows that ATAC-seq and H3K27ac peaks are reduced, but to a lesser degree relative to the reduction of TADs and loops. This would suggest that some of 3D architectural changes do not directly correlate with chromatin accessibility and or enhancer configuration (relevant to

transcription). Again, it will be helpful if the authors provide the meaning of these observations.

We thank this reviewer for addressing this important point. The charts in Figure 4d provide counts of detected features, however, they provide little information regarding quantitative changes within these features. We regret that this may not have been clear enough. We address differences using statistical testing, and changes are summarized in MDS charts provided across the manuscript, and the missing chart for TADs and loop domains are now added in Figure 5e. Nevertheless, this reviewer correctly points out that we see relatively few changes in accessible chromatin, and we regret that we didn't provide appropriate explanations for this finding, which are now added in the revised manuscript.

For sites of open chromatin, the small changes may relate to the fact that the mechanisms that are responsible for creating them (binding of TF and recruitment of remodelling machines) are likely upstream (or perhaps even independent) of cohesin-loading/loop formation. In line with this, most regions that loose accessibility upon CTCF knockdown actually contain the CTCF motif, suggesting that CTCF is directly required for maintaining accessibility at these sites. These sites are also enriched at weakened loop anchors, suggesting that CTCF is required for both maintaining the open chromatin and the loop. However, when we knock-down cohesin, we lose accessible sites containing TF motifs associated with differentiation, but much fewer than we lose loop interactions, and these sites are not enriched at loop anchors.

Enhancer activation may be more sensitive to cohesin, since contacts between regulatory elements may influence the local deposition/removal of histone modifications, by shared activity of co-factors. Hence, we have now extended our analyses of H3K27ac changes. We also added more explanations and possible interpretations of the changes that we observe, hoping that the data makes more sense now.

4. The example of the LPL locus in Fig. 4h is beautiful, but this is only one aspect of the complex changes occurring in Hi-C landscapes. It might be more revealing to characterize TADs and loops linked to the reduction on ATAC-seq and H3K27ac. Are TADs/loops associated with reduced ATAC peaks overlap with those for H3 K27ac? Are the peaks specific for MAC or common with MO? What are ATAC peaks and H3K27ac peaks unaffected by RAD21KD? Are they located within or outside of . TADs/loops?

We thank this reviewer for the suggestions. Overall, the analyses suggested by this reviewer are addressed using gene or peak set enrichment analyses provided in Fig. 5-7. We have incorporated comparisons with MO and provide evidence that genes affected by cohesin-loss are enriched for genes that are regulated during differentiation. As explained above, ATAC is not very informative, because changes are related to either CTCF loss, or to changes in the transcription factor network that are likely secondary to cohesin loss. One also needs to consider that the loops we measure are in most cases the "endpoints" of the loop extrusion process, where cohesin finally bounces into CTCF. A large majority of the loop anchors detected are not regulatory, but structural, and due to their boundary/insulation function their impact on neighboring genes can be positive or negative. Regulatory loops that are created during loop extrusion are mostly transient and sparse, which is why we don't detect them frequently with HiC. Nevertheless, we now also report the overlaps of enhancers and accessible sites with loop anchors (new Supplementary Tables 1-4) and discuss their relationships in more detail in the results section. We hope that the novel analyses and explanations help to clarify the observed effects of cohesin/CTCF depletion.

5. Fig. 7. The data in this figure are important and interesting, since they show that RAD21KD significantly alters LPS induced transcriptome profiles in MAC. The results suggest that the LPS signal changes 3D landscape within 4 h to initiate timely innate immune

responses. However, it is somewhat disconcerting that no Hi-C data are presented for LPS stimulated MAC. Fig. 7 depicts changes in RNA-se profiles in control and RAD21 KD after LPS treatment. It is imperative to present (a) increase/decrease in TADs/loops after LPS stimulation, (b) the relation to ATAC/H3K27ac peaks (as in Fig 3, and Fig. 4) and (c) the effects of RAD21 KD). A more extensive correlation study may be warranted to relate LPS induction (repression) of gene expression with LPS induced changes in Hi-C. I would emphasize that elucidating the functional role of Hi-C conformation is highly consequential, because LPS induces inflammatory responses not only in human macrophages studied here, but widely in myeloid cells of many other species and LPS-induced inflammation is relevant to various diseases.

We thank this reviewer for raising this interesting point. However, we think that this comment is based on a misunderstanding, which we may have caused due to inaccurate discussion of the matter. We did not aim at addressing the certainly interesting question, whether the differential response in control vs. cohesin-depleted macrophages is due to LPS changing the 3D landscape within 4h. It is quite possible that regulatory loops are changed rapidly during LPS activation. But to study this type of biology, we would require a culture system that allows the (very) rapid depletion of RAD21 in monocyte-derived macrophages (to study otherwise mostly equivalent cells), which is not feasible, unfortunately. Also, these days, one would choose a method that is better suited to detect regulatory loops, like microC-based promoter/enhancer capture for example. In Figure 8, we are comparing monocyte-derived macrophages that were either treated (already at the MO stage) with RAD21 siRNA or mock-treated, just like in previous figures and as explained in Fig. 8a. So essentially, the LPS signal is triggered in macrophages with different 3D chromatin conformations (which were analysed in Figures 5-7). Hence, the differential response likely results (at least to some extent) from different 3D structures in the unstimulated (0h) RAD21 siRNA and mock-treated macrophages. We have further clarified this in our revised discussion. Our setting may correspond to a comparison of LPS responses in monocyte-derived macrophages in individuals/patients without or with cohesin mutations in their stem cell compartment.

6. Assuming that LPS stimulation induces changes in chromatin conformation, the authors should comment on possible mechanisms by which LPS treatment can change chromatin architecture so rapidly.

As outlined above, we cannot distinguish, whether differential responses in control vs. cohesin-depleted macrophages are due to LPS rapidly changing the 3D landscape, or due to the altered “3D ground-state” of unstimulated (0h) RAD21 siRNA and mock-treated macrophages. Studying the impact of LPS on chromatin conformation would clearly be interesting, but required a different experimental setup.

7. LPS induction of inflammatory genes highly transient, in that transcription ceases within 24 h for the most part. Are LPS induced TADs/loops similarly transient?

As outlined above, we agree that this is an interesting question. However, we sincerely hope that this reviewer appreciates the wealth of data and analyses provided, and concurs, that answering this question is beyond the scope of our current manuscript.

Reviewer #3 (Remarks to the Author):

This study presents a comprehensive analysis of the chromatin changes during human monocyte differentiation into macrophages and dendritic cells as well as assess the effects of Cohesin and CTCF depletion on this developmental progression. The key findings in this study in human monocytes reflect observations previously seen of cohesin's role in regulation of cell-type specific gene expression, its importance in the differentiation in mouse monocytes and human HSPCs as well as in the regulation of immune response genes.

I have some major comments below I believe are important to provide a complete study.

We would like to thank this reviewer for his/her thoughtful and thorough review and generally helpful comments.

Major points

1. To assess the identity of derived cell-lines and with the availability of RNA-seq for these samples, a transcriptome wide comparison can be made. Markers of these cell lines can be derived from public databases and used as gene sets for a GSEA style of analysis against for the MO, MAC, moDC transcriptomes. This is especially important as the cell identity is central to many of the findings.
 - 1.1 In addition a supplementary heat map of these markers would provide a useful visual overview of the above and provide some understanding of the within sample variability.

We thank the reviewer for highlighting this important point and we totally agree that the definition of cell identity is important. We originally provided mRNA expression levels of typical marker genes as displayed in Fig. 1c and MDS plots in Fig. 1b, indicating that replicates behave similar and regulate known marker genes. To address this further we now also provide a heatmap of differentially expressed genes (as suggested by this reviewer) with additional marker genes highlighted (new Supplementary Figure 1a). This reviewer also suggested to compile a marker gene set from public databases for GSEA-type of analyses. In theory, this is an excellent idea, however, in practice, there is no "consensus" for MAC or DC cultures, because every lab uses its own protocol for generating them and especially macrophages come in thousands of flavors. Hence, we think that it is beyond the scope of this manuscript to curate a consensus marker set for these cell types. As an alternative, we looked for the top gene sets in the ImmuneSigDB subset of the MSigDB (Gene sets representing chemical and genetic perturbations of the immune system generated by manual curation of published studies in human and mouse immunology) that were enriched across MO/moDC and MO/MAC comparisons and show the corresponding barcode plots in the supplement. Both correspond to gene sets from an independent study of MO, MAC and DCs (using similar but not identical culture conditions), confirming our cell type identities. We hope that the revised presentation clarifies the cellular identities of our monocyte-derived cell types.

2. Global changes in genome wide accessibility are expected both over differentiation and , perhaps to a lesser extent, following cohesin depletion. The FRIP scores for ATAC-seq presented in supplementary tables as well as the ATAC-seq signal plots in figure 1 show a confounding of overall ATAC-seq signal with MO, MAC and moDC groups. The use of quantile normalisation in the identification of statistical differences across groups would assume much of this change in efficiency is technical whereas in this context it may be a reflection of expected biology. A similar change in total RNA expression levels may be expected in conjunction with this change in overall accessibility. Studies like this have used external RNA spikes in for RNA-seq and normalisations to total mapped reads for ATACseq.
 - 2.1 The changes on global ATAC efficiency and their impact of differentially accessible sites should be evaluated.

2.2 For the RNA-seq, comments on use of no-spike in comparison to previous studies can be made.

We thank the reviewer for raising two very relevant issues. In principle, we agree that spike-RNAs are helpful to normalize RNAseq data sets of cell types with large changes in total RNA content/cell. However, to use them appropriately, one has to determine the exact cell counts, to match spike-in controls with cell numbers. In our hands, only FACS sorting is accurate enough to reliably make use of spike-ins for normalization purposes. For this particular study, we grew several tens of millions of cells for an array of assays (with at least seven different settings per donor and cell type). Sorting these cells would have taken several hours and was therefore considered unfeasible. Unfortunately, we are also not aware of any previous studies using spike-in controls for RNAseq experiments in non-proliferating monocyte-differentiation system, so it is hard to comment on this. But since we agree that most of our analyses would have benefited from internal controls (including RNAseq, ChIP, ATAC and HiC), we note this as a potential limitation of our study in the discussion.

The comment on ATAC seq normalization, as we understand it, seems to be based on two misunderstandings. First of all, the histograms in Figure 1d do not imply “a confounding of overall ATAC-seq signal with MO, MAC and moDC groups”, since we are looking at sites that were selected based on their differential H3K27ac status and not on differential accessibility – this Figure just tells us that the loss of H3K27ac in MO does not (in average) result in the loss of accessibility, while sites that are specifically gaining H3K27ac during differentiation are often also *de novo* remodelled. In response to this reviewer’s comment, we now also perform motif analyses across differential ATAC peaks, as shown in Supplementary Fig. 1F.

The second misunderstanding relates to the normalization procedure, which is not a “classical” quantile normalization. We apologize for this unnecessary confusion, which was most likely caused by a corresponding statement in the methods section. We have been testing various analysis strategies and the one that we are using here basically performs a quantile regression (not quantile normalization in the classical sense) on the top 5% of peaks, assuming that the fraction of most accessible regions can be reliably compared. While this would not be valid for RNA seq data (where a cell can produce a wide range of transcriptional outputs from a single allele), we feel that this type of assumption is valid for chromatin accessibility, which has a clearly defined range (a single allele at a particular location can either be accessible or not). So, we are basically normalizing our ATAC seq data to signals in peaks for which we assume that they are accessible in most cells. In our experience, this has been the most robust way to treat replicates with variable background. We know that we have technical variance across donors and treatments, as can be seen from variations in FRIP scores of the same cell type across donors. And while FRIP is a QC measure that we use, we always visually inspect the quality of individual coverage tracks. The suggested normalization on total counts (we guess that this reviewer refers to all reads recovered as opposed to reads in peaks only) would be problematic, because some replicates are too variable across replicates to detect significantly different regions. Another reason for using this strategy is that it allows us to account for effects of region length and local GC content.

In our experience ATAC-seq can be very sensitive towards technical noise, and this can vary across donors and sample type. We don’t think that the background reflects expected biology, as suggested by the reviewer. As obvious from the differential accessibility analyses, we observe abundant differences between cell types and upon knockdown, which we expect (e.g. the loss of accessible sites with CTCF motifs upon CTCF knockdown). We hope that we have sufficiently explained our approach and addressed all reviewers concerns.

3. The identification of MO-specific loops which are CTCF independent suggests that another factor is perhaps acting at these sites. This is very interesting observation alongside previous work on de-novo loop formation in other systems. Previous work has identified

monocyte to macrophage, esc to npc and epidermal differentiation loop formation may be less dependent on CTCF binding and associated with specific transcription factors.

- 3.1 In this context, an analysis of the motifs and putative transcription factors at these sites would be important to this study.
- 3.2 A comment on the loss of cohesin independent interactions in respect to whether these observations match any gain of cohesin independent denovo interactions along development in other cell types.

We would like to thank this reviewer for this suggestion.

Of note, we are not aware of any publication describing that differential loop formation is less dependent on CTCF binding during human primary monocyte to macrophage differentiation (as indicated by this reviewer). We assume that this reviewer refers to a study comparing THP-1 cells with PMA-treated THP-1 cells (by Phanstiel et al., *Molecular Cell*, 2017), which is often used as a model for human macrophage differentiation. THP-1 cells are genetically and epigenetically aberrant leukemia cells, and it is unclear (at least to us), whether the changes during PMA treatment of these cells (which induces both growth-arrest and differentiation) are relevant to the biology of human primary monocytes and macrophages. In contrast to PMA-treated, monocyte/macrophage-like THP-1 cells, primary human macrophages primarily gain structural loops, which are characterized by CTCF motifs with higher motif affinities. The type of anchors that change in MO are less dependent on CTCF (with a lower affinity) and may correspond more to what is observed in the THP-1 model. There is one study using promoter-capture Hi-C across 17 different blood cell types (including MO and MAC by Javierre et al., *Cell*, 2016), which showed that promoter-interacting regions are enriched for regulatory chromatin features, but this assay is tailored to the identification of such interactions and would miss structural loops if they are not promoter-centered. We have now included these studies in the discussion of our results.

We have also performed additional motif analysis on various levels. First we asked, which motifs would distinguish CTCF/RAD21-independent loops from those overlapping either one or both CTCF and Rad21 in individual cell types (new Supplementary Figure 3c). Here, we identified YY1 as the top enriched motif, particularly in MO, which has previously been implicated in CTCF-independent looping.

We also looked into CTCF-independent loops (Figure 4 and Supplementary Figure 4), and see that typical enhancer associated TF motifs are enriched in CTCF independent loop anchors. We do not see a single dominant factor here (as opposed to the THP-1 model). Our data suggests that CTCF-independent loops are mostly formed by enhancers/promoters containing the typical signature motifs of MO, MAC or moDC. However, in contrast to the cell line model, we clearly observe a gain of structural loop domains. The overall paucity of regulatory loops likely is a problem of sensitivity of our Hi-C approach.

This is now also discussed in the context of published work. We also included comments on ESC differentiation models.

4. Some more information on what the batches used in correction for visualisation were should be included in the methods. I am assuming patient in this case but this should be stated clearly as additional batch correction on unknown factors is often a problem for reproducibility.

We agree with this reviewer that reproducibility is an important issue. We provide information on batch correction in the methods section. All statistical analyses were done correcting for donor effects. In visualizations of RNA-seq data were corrected for donor & siRNA-specific effects. Hi-C, ChIP- and ATAC-seq data were not batch-corrected for visualizations. Here we generally plotted mean coverage across donors, as now also stated in the Methods.

5. The effects on regulation of inflammation induced genes in cohesin depleted cells is interesting although, as the author notes, opposed to that observed in mouse monocytes. In order to better align the comparisons, a side by side heat map of the induced genes and gene sets from the Cuartero study (translated to human orthologues for the mouse data) would be very informative.

We would like to thank this reviewer for this suggestion. We have now implemented the comparison in the Supplementary Figure 8. We downloaded and “reanalysed” the mouse data (we could reproduce the original findings) and show the species comparisons for both types of regulated genes (either cohesin-dependent in human or in mouse). The differences may not be surprising and are in line with previous studies comparing macrophage responses to LPS between human and mouse. A corresponding comment has been added in the discussion.

6. Given the availability of H3K27ac in this study and since both the connectivity between super enhancers have been shown to be promoted by cohesin, and super enhancers are believed to play an important role in cell identity, I would be interested to hear comment on any effects on SEs over development and following cohesin depletion.

As suggested, we have now included analyses of super-enhancers. In our model, we do see differentiation-associated changes in super enhancers (SE), in particular during differentiation. Here, mostly SE associated with genes characterizing cellular functions are affected. Examples are the CIITA transcription factor, which drives the antigen-presentation program in moDC, or the Natural resistance-associated macrophage protein 1 (encoded by *SLC11A1*), which is involved in iron metabolism and host resistance to certain pathogens in MAC (also see new Supplementary Fig. 1g). Cohesin-depletion did not result in fundamental changes in loop domains overlapping super-enhancers, hence we didn’t follow up on this further.

Notably, the described model captures differentiation at a different scale compared to e.g. ESC-based models. Our starting cell (human blood monocyte) is already a mature effector cell that does not further proliferate. Key lineage drivers (like *Spi1*) are already expressed and operating, and we see genes related to the specific functions of MO-derived cells to be altered during differentiation. This is different from models that undergo differentiation and finally growth arrest starting from stem cells. In the latter systems, changes in lineage-driving genes and genes controlling the proliferative states are frequent (and usually highlighted).

7. Due to the complexity of bioinformatics analyses performed in this study, it is important to provide some processed results.
 - 7.1 --Tables of differential analysis of ATAC, RNA and ChIP should be included.
 - 7.2 --Tables of files of loop boundaries and the overlap with CTCF/Cohesin should be included

We appreciate this point, and regret that we did not provide these files in the initial submission. The requested tables are now available as Supplementary data files 1-4. Additional data is provided in the Source data file.

A few minor comments.

8. It would be interesting to see the enriched motifs represented as a motif by sample heat map (such as implemented in ChromVar). This would allow for a visual assessment of the

relative strength of enrichment for motifs across groups and the variability of enrichment within samples.

We thank this reviewer for suggesting the use of ChromVar, which uses a different approach based on TF footprints. We do see the potential benefit of adding ChromVar analyses and will consider its usage in future analyses, however, since we shift fragments early during our analyses (at the level of BAM files), ChromVar is currently not compatible with our analysis pipeline. We mostly rely on *de novo* motif identification (which is a different approach). This has been extremely powerful in identifying motifs across peaks or open chromatin regions which are selected based on statistical models that take replicates into account already. The selection of known motif families are then annotated based on the *de novo* results to be able to compare motif compositions across samples. We prefer the balloon plot presentation over heatmaps of motif enrichment, because it visualizes both enrichment and significance (heatmaps only show enrichment).

9. Figure 2c could be presented as epigenetic heat map (as in 3f) of all the loop domains groups as illustrated in the the Figure 2c venn diagram to provide more resolution in the correlation of absolute ChIP signals to reported overlaps.

We thank this reviewer for his/her suggestion and agree that genomic distance plots or epigenetic heatmaps provide much more resolution. We have now replaced or added these types of presentations where appropriate.

10. Figures 2g and f, it is unclear what the multiple correction applied is and whether it is effected by how many tests were conducted at the same time (as in BH or bonferroni).

We thank the reviewer for indicating the lack clarity in our description of the multiple correction method applied. Gene or peak set testing normally included two test for each setting, so each P value is adjusted for two tests using the BH correction implemented in fry (limma). This information has also been added in the text.

11. “Genome ontology analysis” - I am unclear what this is and couldn't find the description in methods.

This term has been used to describe a test for enrichment of particular genomic features (in analogy to functional features in gene ontology analyses). We have replaced the term and now provide a more comprehensible description of the analysis.

12. Figure 2i is missing what the enrichment value is on y axis.

The missing values were added in the revised manuscript.

13. Supplementary Figure 2i should point is now supplementary figure 2f I believe.

The reference to this Figure has now been corrected.

14. Venn diagrams of overlaps (2c/3b) should be presented as tables in supplementary alongside odds ratios/p-values.

The Venn diagram in 2c was entirely replaced by genomic distance plots, as suggested by the reviewer. Same for 3b (now Fig. 4a), were we kept the diagrams because we feel that this type of visualization makes it easier to capture the differences in overlaps.

15. “A large fraction of MO-specific loops lacked CTCF binding and instead were marked by H3K27ac, which was rarely seen in MAC or moDC-specific loops” should only point to Figures 3f and Supplementary Figures 3g-h.

The references have been corrected in the new version.

16. The observation of differential lengths for chromatin loops following cohesin depletion in MAC and moDC is interesting and in keeping with previous finding. Is there any potential mechanism why CTCF depletion shows a similar effect in MAC but not moDC cells?

We thank the reviewer for raising this point. Actually, we had observed several smaller inconsistencies in the analyses of CTCF knockdown experiments between MAC and moDC, which prompted us to change the normalization strategy for RAD21 and CTCF knockdown samples. In the revised version, we analysed differential loops using an approach that normalizes on total interactions. We think that since we preferentially loose interactions in both settings, this type of normalization is more appropriate for the knock-downs. With the revised procedure, MAC and moDC now show similar distributions. We assume that this is more or less an effect of a.) reducing CTCF and b.) having both more and less resistant CTCF sites. This would naturally lead to the observed differences in loop sizes. We now also comment on this in the results section of the manuscript.

17. A supplementary epigenetics heat map to complement figure 4h showing CTCF and cohesin signal over strengthened and lost anchors would allow a reader to better assess the strength of peaks that overlap at these sites.

This section has been completely redone, and we now include the suggested epigenetic heatmaps as well motif score analyses, suggesting that anchors of weakened loops contain lower affinity CTCF binding sites.

18. I think Figure 5C would better fit a Chromvar heat map to show distribution of motif enrichment between conditions/cell types and variability of motif enrichment within samples.

As stated above, we currently prefer the ballon plot representation. We do see the potential benefit of adding ChromVar analyses and will consider its usage in future analyses, however, since we shift fragments early during our analyses (at the level of BAM files), ChromVar is currently not compatible with our analysis pipeline.

19. “Except for motif signatures (likely due to a higher background in moDC ATAC-seq data)” — How is higher back ground determined? From 1-FRIP?

Yes. We use the FRIP values determined by HOMER. We see fewer reads in peaks in some moDC samples.

20. It is not clear to me how figure 5f and 6c were derived “KEGG pathway analysis of genes affected by CTCF loss in MAC”.

We used the kegg function provided in the limma package to identify KEGG pathways enriched in our differentially expressed gene. This was already stated in the Methods section and may have been overlooked by the reviewer.

21. Why two different tests are needed for ChiP differential analysis, (for regions and peaks) is unclear to me.

Peaks have fixed sizes and regions have variable sizes. In the second case, we account for length differences in our analyses.

22. How are peaks defined for ATACseq/ChIPseq differential analysis over peaks? Were they reduced/collapsed into representative peaks such as implemented in DiffBind.

We apologize for the lack of clarity in our initial submission. Peak sets are initially defined using combined reads from individual settings (e.g. all MO samples) and merged for differential analyses (e.g. MO peaks and MAC peaks). We use the mergePeaks program provided in the HOMER package for merging. We now added this missing information in the methods section.

23. The exact version of the GRCh38 genome used and a link to its source should be included in the supplementary for ease of reproducibility.

We appreciate this comment and include this information in the Methods section of the revised manuscript. All analyses were done using GRCh38.p10, the link to GENCODE has been added (https://www.gencodegenes.org/human/release_27.html).

24. The description in the text of the removal of multmapping reads after Bowtie2 alignment should be altered. Filtering by mapQ > 10 can include multi mapping reads and instead a strategy employing the filtering by XS SAM tag could be used. The text could be changed to "removal of lower quality alignments by filtering reads with mapQ <= 10". In my experience, this will not affect downstream analysis as mapQ reflects a scale of uniqueness in Bowtie2 alignment and removal of > 10 will indeed remove major artefacts but clarifying the methods for others wishing to follow is useful here.

We apologize for the lack of clarity in our initial submission. In essence, both the filtering for multi mapping reads and mapq filtering is done when we generate HOMER tagDirectories. The text has been corrected accordingly.

25. removeBatchEffect is a function in the Limma package, not edgeR.

We corrected the reference.

26. A supplementary heatmap showing the correlation between replicate ATAC-seq signals before and after batch correction would be helpful for the reader to assess inter patient variability and variability across replicates post batch-correction.

In response to this comment, we would like to point out that this manuscript focuses on chromatin conformation and primarily analyses other features (like gene expression, chromatin accessibility, or H3K27ac deposition) in this context. We agree that donor variability is potentially an interesting issue, but since we only have three donors, any analyses of donor variability are clearly underpowered. Also, the suggested analyses should be done for gene expression, chromatin accessibility, H3K27ac deposition etc. which would blow up our supplemental data entirely and distract readers.

27. "Statistically significant differences in read counts across peaks between sets of replicate ATAC-seq experiments were determined with quantile (0.95) normalization and GC correction using edgeR (v3.32.1) with the cqn package in R (4.0.3)". I am unsure if the replicates were used for this analysis or averaged for this test.

All differential analyses throughout the manuscript were performed using individual replicates as stated.

REVIEWERS' COMMENTS

Reviewer #2 (Remarks to the Author):

The authors have made Large changes in this revised manuscript, which substantially enhanced the clarity and readability.

This reviewer offered some comments on the last section of the Results section on "Innate activation programs are altered (line (line 461)). The authors replied stating that "We did not aim at addressing the certainly interesting question, whether the differential response in control vs. cohesin-depleted macrophages is due to LPS changing the 3D landscape within 4h". However, my question is somewhat more general, including whether and how LPS treatment alters the 3D architecture of the wild type (WT) MAC. In my view, this question is part of the central thrust of this manuscript, integrating biological phenotypes, transcriptome programs and Hi-C profiles. So, the data on Hi-C would have revealed the relation between LPS treatment and RNA-seq data/ morphological changes. The question indeed precedes the issue of comparing WT and KD cells, which the authors emphasize the technical difficulty. Nevertheless, I understand that the authors do not wish to go into this issue further in this manuscript (perhaps presenting the data elsewhere). Thus, this reviewer would like to simply suggest that the authors comment on the question of if and how innate immune stimulations, such as LPS (poly:IC and other pathogen components) would affect 3D conformation in WT myeloid cells in an appropriate section of the paper.

Reviewer #3 (Remarks to the Author):

The manuscript is much improved and the authors have addressed my comments and provided the additional information asked for.

Response to reviewer's comments:

We would like to thank the reviewers for the time and effort they have invested in reviewing our manuscript. One reviewer made one additional remark, which we have now addressed in the revised manuscript:

Reviewer 2 comment:

The authors have made Large changes in this revised manuscript, which substantially enhanced the clarity and readability.

This reviewer offered some comments on the last section of the Results section on “Innate activation programs are altered (line (line 461). The authors replied stating that “We did not aim at addressing the certainly interesting question, whether the differential response in control vs. cohesin-depleted macrophages is due to LPS changing the 3D landscape within 4h”. However, my question is somewhat more general, including whether and how LPS treatment alters the 3D architecture of the wild type (WT) MAC. In my view, this question is part of the central thrust of this manuscript, integrating biological phenotypes, transcriptome programs and Hi-C profiles. So, the data on Hi-C would have revealed the relation between LPS treatment and RNA-seq data/ morphological changes. The question indeed precedes the issue of comparing WT and KD cells, which the authors emphasize the technical difficulty. Nevertheless, I understand that the authors do not wish to go into this issue further in this manuscript (perhaps presenting the data elsewhere). Thus, this reviewer would like to simply suggest that the authors comment on the question of if and how innate immune stimulations, such as LPS (poly:IC and other pathogen components) would affect 3D conformation in WT myeloid cells in an appropriate section of the paper.

We agree that the questions of whether and how innate stimulations induce conformational changes in macrophages and which role cohesin plays in regulating the transcriptional response are indeed very interesting. We had already included some discussion in the previous version of the manuscript and extended on this a little bit further, also providing some references related to this topic. The discussion now reads:

“Along the same lines, we also show that cohesin-depleted MAC display a strongly altered innate immune activation profile. Here, genes induced by the inflammatory stimulus may be affected on various levels: 1.) Their response may be affected by the altered chromatin conformation and interaction landscapes in MAC that were differentiated in the presence or absence of cohesin. 2.) Their response may also be altered due to secondary alterations in signaling networks required to maintain the normal response. 3.) Their response may be directly affected by cohesin-depletion, which **may prevent** the formation of LPS-induced loops that are necessary for a normal response. **Interaction landscapes can rapidly change during macrophage stimulation, as exemplified by conformation changes observed after 2h of IFN γ stimulation in mouse macrophages⁵⁷ or 2h of LPS treatment of THP1 cells⁵⁸. NF κ B and AP-1, the main TF driving the early responses to pathogen-associated molecular patterns like LPS, frequently bind promoter-distal sites⁵⁹ and may require cohesin to properly interact with their target promoters during activation.**“